

# Application of ensemble transform data assimilation methods for parameter estimation in nonlinear problems

Sangeetika Ruchi[1] and Svetlana Dubinkina[1]

[1]Centrum Wiskunde & Informatica, P.O. Box 94079, 1098 XG Amsterdam, The Netherlands

**Correspondence:** Sangeetika Ruchi (s.ruchi@cwi.nl)

**Abstract.** Over the years data assimilation methods have been developed to obtain estimations of uncertain model parameters by taking into account a few observations of a model state. However, most of these computationally affordable methods have assumptions of Gaussianity, e.g. an Ensemble Kalman Filter. Ensemble Transform Particle Filter does not have the assumption of Gaussianity and has proven to be highly beneficial for an initial condition estimation and a small number of parameter

estimation in chaotic dynamical systems with non-Gaussian distributions. In this paper we employ Ensemble Transform Particle Smoother (ETPS) and Ensemble Transform Kalman Smoother (ETKS) for parameter estimation in nonlinear problems with 1, 5, and 2500 uncertain parameters and compare them to importance sampling (IS). We prove that the updated parameters obtained by ETPS lie within the range of an initial ensemble, which is not the case for ETKS. We examine the performance of ETPS and ETKS in a twin experiment setup, where observations of pressure are synthetically created based on the know

values of parameters. The numerical experiments demonstrate that the ETKS provides good estimations of the mean parameters but not of the posterior distributions and as the ensemble size increases the posterior does not improve. ETPS provides good approximations of the posterior and as the ensemble size increases the posterior converges to the posterior obtained by IS with a large ensemble. ETKS is very robust while ETPS is very sensitive with respect to the initial ensemble. An issue of an increase in the root mean square error after data assimilation is performed in ETPS for a high-dimensional test problem is resolved by

applying distance-based localization, which however deteriorated the posterior estimation.

## 1   Introduction

An accurate estimation of subsurface geological properties like permeability, porosity etc. is essential for many fields specially where such predictions can have large economic or environmental impact, for instance prediction of oil or gas reservoir locations. Knowing the geological parameters a so-called forward model is solved for the model state and a prediction can be made.

The subsurface reservoirs, however, are buried thousands of feet below the earth surface and exhibit a highly heterogeneous structure, which makes it difficult to obtain their geological parameters. Usually a prior information about the parameters is given, which still needs to be corrected by observations of pressure and production rates. These observations are, however, known only at well locations that are often hundreds of meter apart and corrupted by errors. This gives instead of a well-posed forward problem an ill-posed inverse problem of estimating uncertain parameters, since many possible combinations of

parameters can result in equally good matches to the observations.



Different inverse problem approaches for groundwater and petroleum reservoir modelling, generally termed as history matching, have been developed over the past years, e.g. Oliver et al. (1997) implemented Markov chain Monte Carlo methods with different perturbations and tested it on a 2-D reservoir model; Reynolds et al. (1996) obtained reservoir parameters estimations using Gauss-Newton method; Vefring et al. (2006) used Levenberg–Marquardt method to characterize reservoir pore

pressure and permeability. A review of history matching developments is written by Oliver and Chen (2011).

For reservoir models the term data assimilation and history matching are used interchangeably, as the goal of data assimilation is the same as that of history matching, where observations are used to improve a solution of a model. Ensemble data assimilation methods such as Ensemble Kalman filters (Evensen, 2009) have been originally developed in meteorology and oceanography for the state estimation. Now it is one of the frequently employed approaches for parameter estimation in subsur-

face flow models as well (e.g. Oliver et al., 2008). The model state is augmented with uncertain parameters and the correlations between uncertain parameters and predicted data are used to correct the parameters together with the state. The simultaneous update of the model state and parameters, however, results in a unbalanced (unphysical) model state. Therefore Evensen (2000) introduced an ensemble smoother, where only uncertain parameters are estimated and the model state is computed by solving the forward model with corrected parameters. A detailed review of ensemble Kalman filter developments in reservoir engineer-

ing is written by Aanonsen et al. (2009). An ensemble Kalman filter efficiently approximates a true posterior distribution if the distribution is not far from Gaussian, as it corrects only the mean and the variance. For nonlinear models with multimodal distributions, however, an ensemble Kalman filter fails to correctly estimate the posterior, as shown by Dovera and Della Rossa (2011).

Particle filtering (Doucet et al., 2001), also known as Importance Sampling (IS), is quite promising for such models as it

does not have any assumptions of Gaussianity. It is also an ensemble based method in which the probability density function is represented by a number of particles (also called samples or ensemble members). One particle corresponds to one configuration of uncertain model parameters. The forward model is solved for each particle and predicted data is computed. The weight is assigned to particles based on the observations of the true physical system and the predicted data. A significant work for parameter estimation using particle filtering has been done in hydrology. Moradkhani et al. (2005) used it to estimate model

parameters and state posterior distributions for a rainfall-runoff model. Weerts and El Serafy (2006) compared an ensemble Kalman filter and a particle filter with different resampling strategies for a rainfall-runoff forecast and obtained that as the number of particles increases the particle filter outperforms the ensemble Kalman filter. Guingla et al. (2012) employed particle filtering to correct the soil moisture and to estimate hydraulic parameters.

Particle filtering, however, does not update the uncertain parameters only their weight. Ensemble transform particle filter

developed by Reich and Cotter (2015) is a particle filtering method that resamples the particles based on their weights and covariance maximization among the particles. Therefore it has an IS advantage of predicting the correct posterior but does not have its disadvantage of resampling lacking. Ensemble transform particle filter has been used for initial condition estimations and for parameter estimations in chaotic dynamical systems with a small number of uncertain parameters (Lorenz 63 model). It has not been applied, however, in subsurface reservoir modelling for estimating a large number of uncertain parameters. In

this paper we employ it for estimating uncertain parameters in subsurface reservoir modelling. We call it ensemble transform





particle smoother (ETPS) in analogy with ensemble smoothers, where the model state is computed from the forward model with corrected model parameters. ETPS does not use correlations between predicted data and uncertain parameters. On the one hand, the correlations should improve the estimation but on the other hand, it is not always clear how to compute them. Ensemble Transform Kalman Smoother (ETKS) developed by Bishop et al. (2001) also does not employ the correlations in

its estimation. It transforms the state from the model space to the ensemble space, minimises the uncertainty in the ensemble space and transforms the estimation back to the model space.

In this paper we investigate the performance of ETPS and ETKS for parameter estimation in nonlinear problems and compare them to IS with a large ensemble. This paper is organized as follows: in the section 2 we describe IS, ETPS, and ETKS for parameter estimation. We apply these methods in Sect. 3 to a one parameter nonlinear test case, where the posterior can be

computed analytically, and in Sect. 4 to a single-phase Darcy flow, where the number of parameters is 5 and 2500. In Sect. 5 we draw the conclusions.

## 2   Data assimilation methods

We implement an ensemble transform Kalman smoother and an ensemble transform particle smoother for estimating parameters of subsurface flow. Both of these methods are based on Bayesian framework. Assume we have an ensemble of $M$

model parameters $\{\boldsymbol{u}_m\}_{m=1}^M$, then according to this framework, the posterior distribution, which is the probability distribution $\pi(\boldsymbol{u}_m|\boldsymbol{y}_{\mathrm{obs}})$ of the model parameters $\boldsymbol{u}_m$ given a set of observations $\boldsymbol{y}_{\mathrm{obs}}$, can be estimated by the pointwise multiplication of the prior probability distribution $\pi(\boldsymbol{u}_m)$ of the model parameters $\boldsymbol{u}_m$ and the conditional probability distribution $\pi(\boldsymbol{y}_{\mathrm{obs}}|\boldsymbol{u}_m)$ of the observations given the model parameters, which is also referred as the likelihood function,

$$\pi(\boldsymbol{u}_m|\boldsymbol{y}_{\mathrm{obs}}) = \frac{\pi(\boldsymbol{y}_{\mathrm{obs}}|\boldsymbol{u}_m)\pi(\boldsymbol{u}_m)}{\pi(\boldsymbol{y}_{\mathrm{obs}})}.$$

The denominator $\pi(\boldsymbol{y}_{\mathrm{obs}})$ represents the marginal of observations and can be expressed as:

$$\pi(\boldsymbol{y}_{\mathrm{obs}}) = \sum_{m=1}^M \pi(\boldsymbol{y}_{\mathrm{obs}}, \boldsymbol{u}_m) = \sum_{m=1}^M \pi(\boldsymbol{y}_{\mathrm{obs}}|\boldsymbol{u}_m)\pi(\boldsymbol{u}_m),$$

which shows that $\pi(\boldsymbol{y}_{\mathrm{obs}})$ is just a normalisation factor.

### 2.1   Ensemble Transform Kalman Smoother

We employ an ensemble Kalman smoother based on a transformation of an ensemble from the model phase space to the

ensemble space—the ensemble transform Kalman smoother Bishop et al. (2001).

Assume we have an ensemble of $M$ initial model parameters $\{\boldsymbol{u}_m^b\}_{m=1}^M$, where $b$ refers to a background (prior) ensemble, which are sampled from a chosen prior probability density function, then the ensemble Kalman estimate (or analysis) $\{\boldsymbol{u}_m^a\}_{m=1}^M$ is given by:

$$\boldsymbol{u}_m^a = \sum_{l=1}^M \mathrm{diag}\left(s_{lm} + q_l - \frac{1}{M}\right)\boldsymbol{u}_l^b, \quad m = 1, \ldots, M,$$



where diag is a diagonal matrix, $s_{lm}$ is the $(l,m)$ entry of a matrix $\mathbf{S}$

$$\mathbf{S} = \left[ \mathbf{I} + \frac{1}{M-1}(\mathbf{A}^b)^T \mathbf{R}^{-1} \mathbf{A}^b \right]^{-1/2}, \tag{1}$$

and $q_l$ is the $l$-th entry of a column $\boldsymbol{q}$

$$\boldsymbol{q} = \frac{1}{M-1} \mathbf{1}_M - \mathbf{S}^2 (\mathbf{A}^b)^T \mathbf{R}^{-1} (\bar{\boldsymbol{y}}^b - \boldsymbol{y}_{\text{obs}}).$$

Here $\mathbf{I}$ is an identity matrix of size $M \times M$, $\mathbf{1}_M$ is a vector of size $M$ with all ones, $\bar{\boldsymbol{y}}^b$ is the mean of the predicted data defined by

$$\bar{\boldsymbol{y}}^b = \frac{1}{M} \sum_{m=1}^{M} \boldsymbol{y}_m^b,$$

$\mathbf{A}^b$ is the background ensemble anomalies of the predicted data defined as

$$\mathbf{A}^b = \left[ (\boldsymbol{y}_1^b - \bar{\boldsymbol{y}}^b) \quad (\boldsymbol{y}_2^b - \bar{\boldsymbol{y}}^b) \quad \dots \quad (\boldsymbol{y}_M^b - \bar{\boldsymbol{y}}^b) \right],$$

and $\mathbf{R}$ is the measurement error covariance. To ensure that the anomalies of analysis remain zero centered we check whether $\mathbf{A}^a \mathbf{1} = \mathbf{A}^b \mathbf{S} \mathbf{1} = \mathbf{0}$, given $\mathbf{S} \mathbf{1} = \mathbf{1}$ and $\mathbf{A}^b \mathbf{1}_M = \mathbf{0}$. The model parameters $\boldsymbol{u}_m^b$ and the predicted data $\boldsymbol{y}_m^b$ are related by $\boldsymbol{y}_m^b = h(\boldsymbol{u}_m^b)$, where $h$ is a nonlinear function and here we assume that the function $h$ is known.

## 2.2 Ensemble Transform Particle Smoother

In particle filtering we represent the probability distribution function using ensemble members (also called particles) as in
ensemble Kalman filter. We start by assigning prior (background) weights $\{w_m^b\}_{m=1}^M$ to $M$ particles and then compute new (analysis) weights $\{w_m^a\}_{m=1}^M$ using the Bayes' formula and observations $\boldsymbol{y}_{\text{obs}}$

$$w_m^a = \frac{\pi(\boldsymbol{y}_{\text{obs}}|\boldsymbol{u}_m^b) w_m^b}{\pi(\boldsymbol{y}_{\text{obs}})}. \tag{2}$$

It is important to note that particle filters *do not* change the parameters $\boldsymbol{u}$, they only modify the weight of the particles. Therefore a sophisticated perturbation needs to be implemented for parameter estimation. Instead particle filtering has been
modified using a coupling methodology which resulted in an ensemble transform particle filter Reich and Cotter (2015). Since at the data assimilation step we update only the parameters and not the states, with analogy to ensemble smoothers we will call this method ensemble transform particle smoother (ETPS).

We assume that initially all particles have equal weight, thus $w_m^b = 1/M$ for $m = 1, \dots, M$, and that the likelihood is Gaussian with error covariance matrix $\mathbf{R}$, then from Eq. (2) $w_m^a$ is given by

$$w_m^a = \frac{\exp \left[ -\frac{1}{2} (\boldsymbol{y}_m^b - \boldsymbol{y}_{\text{obs}})^T \mathbf{R}^{-1} (\boldsymbol{y}_m^b - \boldsymbol{y}_{\text{obs}}) \right]}{\sum_{j=1}^M \exp \left[ -\frac{1}{2} (\boldsymbol{y}_j^b - \boldsymbol{y}_{\text{obs}})^T \mathbf{R}^{-1} (\boldsymbol{y}_j^b - \boldsymbol{y}_{\text{obs}}) \right]}, \quad m = 1, \dots, M. \tag{3}$$



In a well-known Importance Sampling (IS) data assimilation method, which will be used in this paper as a "ground" truth, these weights define the posterior pdf. The mean parameter for IS is then

$$\bar{\boldsymbol{u}}^a = \sum_{m=1}^{M} \boldsymbol{u}_m^b w_m^a.$$

ETPS looks for a coupling between two discrete random variables $B_1$ and $B_2$ so as to convert the ensemble members belonging to the random variable $B_2$ with probability distribution $\pi(B_2 = \boldsymbol{u}_m^b) = w_m^a$ to the random variable $B_1$ with uniform probability distribution $\pi(B_1 = \boldsymbol{u}_m^b) = 1/M$. The coupling between these two random variables is an $M \times M$ matrix $\mathbf{T}$ whose entries should satisfy

$$t_{mj} \geq 0, \quad m, j = 1, \ldots, M, \tag{4}$$

$$\sum_{m=1}^{M} t_{mj} = \frac{1}{M}, \quad j = 1, \ldots, M, \tag{5}$$

$$\sum_{j=1}^{M} t_{mj} = w_m^a, \quad m = 1, \ldots, M. \tag{6}$$

An optimal coupling matrix $\mathbf{T}^*$ with elements $t_{mj}^*$ minimizes the squared Euclidean distance

$$J(t_{mj}) = \sum_{m,j=1}^{M} t_{mj} ||\boldsymbol{u}_m^b - \boldsymbol{u}_j^b||^2 \tag{7}$$

and the analysis model parameters are obtained by the linear transformation

$$\boldsymbol{u}_j^a = M \sum_{m=1}^{M} t_{mj}^* \boldsymbol{u}_m^b, \quad j = 1, \ldots, M. \tag{8}$$

Then the mean parameter for ETPS is

$$\bar{\boldsymbol{u}}^a = \sum_{m=1}^{M} \boldsymbol{u}_m^a \frac{1}{M}.$$

We use $FastEMD$ algorithm of Pele and Werman (2009) to solve the linear transport problem and get the optimal transport matrix.

    **Remark:** An important property of ETPS is to retain the imposed interval bounds of ensemble members. Consider an

ensemble of parameters $\{\boldsymbol{u}_m^b\}_{m=1}^{M}$ given by

$$\boldsymbol{u}_m^b = (a_m^b \; b_m^b \; c_m^b)^T, \quad m = 1, \ldots, M,$$

where we assume all the parameters $\{a_m^b\}_{m=1}^{M}$, $\{b_m^b\}_{m=1}^{M}$ and $\{c_m^b\}_{m=1}^{M}$ are bounded between $0$ and $1$. Therefore, the following inequalities hold:

$$0 < a_{\min} \leq a_m^b \leq a_{\max} < 1, \quad m = 1, \ldots, M,$$

$$0 < b_{\min} \leq b_m^b \leq b_{\max} < 1, \quad m = 1, \ldots, M,$$

$$0 < c_{\min} \leq c_m^b \leq c_{\max} < 1, \quad m = 1, \ldots, M.$$





Now we assume two discrete random variables $B_1$ and $B_2$ have probability distributions given by

$$\pi(B_1 = \boldsymbol{u}_m^b) = 1/M, \quad \pi(B_2 = \boldsymbol{u}_m^b) = w_m^a,$$

with $w_m^a \geq 0$, $m = 1, \ldots, M$ and $\sum_{m=1}^M w_m^a = 1$. As ETPS looks for a matrix $\mathbf{T}^*$ which defines coupling between these two probability distributions, each entry of this coupling matrix satisfies the conditions given by Eq. (4)–(6). These conditions assure that each entry of the coupling matrix will be non-negative and less than 1. Since the analysis given by Eq. (8) is

$$\boldsymbol{u}_m^a = \begin{bmatrix} a_1^b(Mt_{1m}^*) + a_2^b(Mt_{2m}^*) + \cdots + a_M^b(Mt_{Mm}^*) \\ b_1^b(Mt_{1m}^*) + b_2^b(Mt_{2m}^*) + \cdots + b_M^b(Mt_{Mm}^*) \\ c_1^b(Mt_{1m}^*) + c_2^b(Mt_{2m}^*) + \cdots + c_M^b(Mt_{Mm}^*) \end{bmatrix}, \quad m = 1, \ldots, M,$$

these conditions lead to

$$0 < a_{\min} \leq a_m^a \leq a_{\max} < 1, \quad m = 1, \ldots, M,$$

$$0 < b_{\min} \leq b_m^a \leq b_{\max} < 1, \quad m = 1, \ldots, M,$$

$$0 < c_{\min} \leq c_m^a \leq c_{\max} < 1, \quad m = 1, \ldots, M.$$

Thus the coupling matrix bounds the analysis ensemble members to be in the desired range. This is not observed in ETKS as the matrix $\mathbf{S}$ given by Eq. (1) does not impose any of the non-equality and equality constraints, so it results in values outside the bound.

## 2.3 Localization

All variations of ensemble Kalman filter and particle filter are limited by the ensemble size. Since, even if the dimension of the problem is just up to a few thousands, a large ensemble size will make each run of the model computationally very expensive. This limit of a small ensemble size introduces a sampling error. To deal with this issue localization for ETKS was introduced by Hunt et al. (2007). We use a distance based localization method. More advanced methods such as wavelet-based approaches of Chen and Oliver (2012) are outside the scope of this paper.

For the local update of a model parameter $\boldsymbol{u}_m(X_i)$ at a grid point $X_i$, we introduce a diagonal matrix $\hat{\mathbf{C}}_i \in R^{N_y \times N_y}$ in the observation space with an element

$$(\hat{\mathbf{C}}_i)_{ll} = \rho\left(\frac{||X_i - r_l||}{r_{\text{loc}}}\right), \tag{9}$$

where $i = 1, \ldots, n^2$, $l = 1, \ldots, N_y$, $n^2$ is the number of model parameters, $N_y$ is the dimension of the observation space, $r_l$ denotes the location of the observation, $r_{\text{loc}}$ is a localisation radius and $\rho(\cdot)$ is a taper function, such as Gaspari-Cohn function Gaspari and Cohn (1999)

$$\rho(r) = \begin{cases} 1 - \frac{5}{3}r^2 + \frac{5}{8}r^3 + \frac{1}{2}r^4 - \frac{1}{4}r^5, & 0 \leq r \leq 1, \\ -\frac{2}{3}r^{-1} + 4 - 5r + \frac{5}{3}r^2 + \frac{5}{8}r^3 - \frac{1}{2}r^4 + \frac{1}{12}r^5, & 1 \leq r \leq 2, \\ 0, & 2 \leq r. \end{cases}$$



Then the estimated model parameter at the location $X_i$ is

$$\boldsymbol{u}_m^a(X_i) = \sum_{l=1}^{M} \mathrm{diag}\left(s_{lm}(X_i) + q_l(X_i) - \frac{1}{M}\right) \boldsymbol{u}_l^b(X_i), \quad m = 1,\dots,M, \tag{10}$$

where diag is a diagonal matrix, $s_{lm}(X_i)$ is the $(l,m)$ entry of the localized transformation matrix $\mathbf{S}(X_i)$

$$\mathbf{S}(X_i) = \left[\mathbf{I} + \frac{1}{M-1}(\mathbf{A}^b)^T(\hat{\mathbf{C}}_i\mathbf{R}^{-1})\mathbf{A}^b\right]^{-1/2}$$

and $q_l(X_i)$ is the $l$-th entry of the localized column $\boldsymbol{q}(X_i)$

$$\boldsymbol{q}(X_i) = \frac{1}{M-1}\mathbf{1}_M - \mathbf{S}(X_i)^2(\mathbf{A}^b)^T\mathbf{R}^{-1}(\bar{\boldsymbol{y}}^b - \boldsymbol{y}_{\text{obs}}).$$

Localization at ETPS introduced by Reich and Cotter (2015) modifies the likelihood and thus the weights given by Eq. (3) are computed locally at each grid $X_i$

$$w_m^a(X_i) = \frac{\exp\left[-\frac{1}{2}(\boldsymbol{y}_m^b - \boldsymbol{y}_{\text{obs}})^T(\hat{\mathbf{C}}_i\mathbf{R}^{-1})(\boldsymbol{y}_m^b - \boldsymbol{y}_{\text{obs}})\right]}{\sum_{j=1}^{M}\exp\left[-\frac{1}{2}(\boldsymbol{y}_j^b - \boldsymbol{y}_{\text{obs}})^T(\hat{\mathbf{C}}_i\mathbf{R}^{-1})(\boldsymbol{y}_j^b - \boldsymbol{y}_{\text{obs}})\right]}, \quad m = 1,\dots,M, \tag{11}$$

where $\hat{\mathbf{C}}_i$ is the diagonal matrix given by Eq. (9). Then the estimated model parameter $\boldsymbol{u}_j^a(X_i)$ at the grid $X_i$ is given by

$$\boldsymbol{u}_j^a(X_i) = M \sum_{m=1}^{M} t_{mj}^* \boldsymbol{u}(X_i)_m^b, \quad j = 1,\dots,M,$$

where $t_{mj}^*$ is an element of an optimal coupling matrix $\mathbf{T}^*$ which minimizes the squared Euclidean distance at the grid point $X_i$

$$J(t_{mj}) = \sum_{m,j=1}^{M} t_{mj}[u_m^b(X_i) - u_j^b(X_i)]^2, \tag{12}$$

which reduces the localized ETPS to a univariate transport problem. It should be noted that localization can be applied only for grid-dependent parameters.

**Remark:** A so-called $R0$ approximation of Reich and Cotter (2015) consists of computing the weights according to Eq. (3) instead of Eq. (11) but solving the optimization problem for each grid (or each parameter) defined by Eq. (12) separately. The $R0$ approximation has an advantage of being a solution of a parallel and computationaly less expensive univariate transport

problem. For parameter estimation of independent parameters as the ensemble size increases the $R0$ approximation defined by Eq. (12) converges to the full approximation defined by Eq. (7) since the sampling noise reduces (not shown). Therefore for computing the posterior distributions with large ensembles we use the $R0$ approximation.

## 3 One parameter nonlinear problem

First we consider a one parameter nonlinear problem from Chen and Oliver (2013). The prior distribution is Gaussian distribu-

tion with mean 4 and variance 1. An observation of a function

$$h(u) = \frac{7}{12}u^3 - \frac{7}{2}u^2 + 8u$$

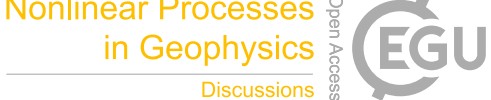



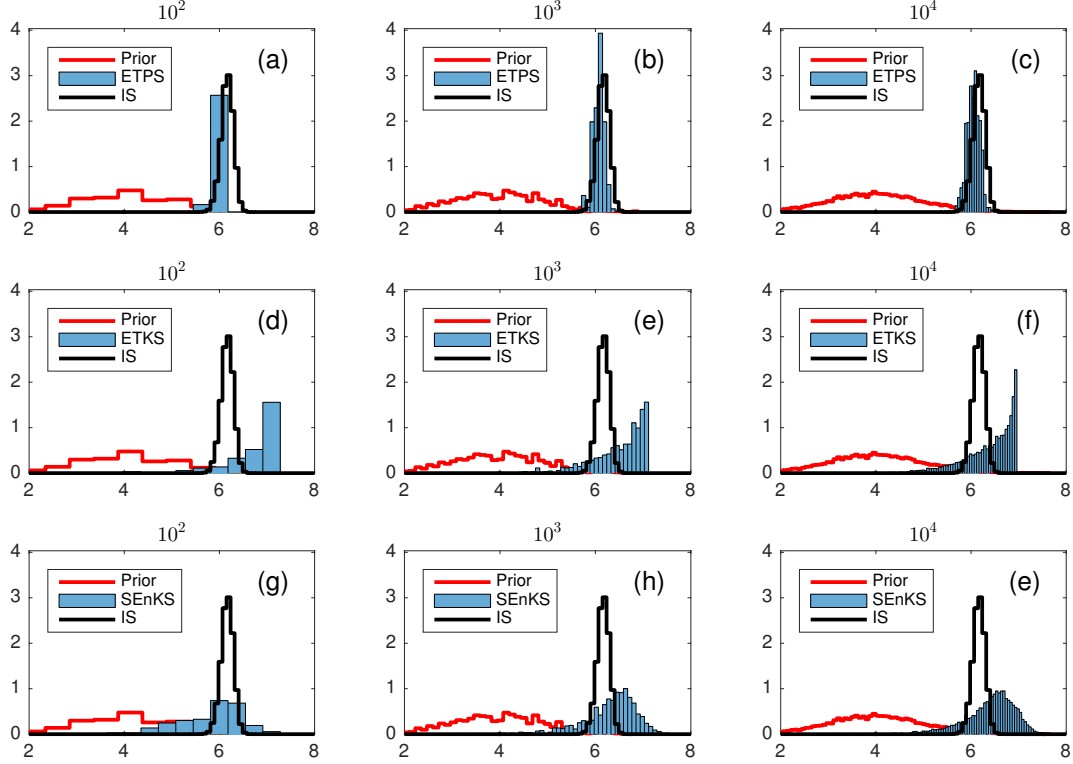

**Figure 1.** Probability density functions for the one parameter nonlinear problem. Top: ETPS, middle: ETKS, bottom: SEnKS. Left: ensemble size $10^2$, center: ensemble size $10^3$, right: ensemble size $10^4$. Prior is in red. True pdf obtained by IS with ensemble size $10^5$ is in black.

is made such that $y_{obs}$ = 48. Observation error is drawn from a Gaussian distribution with zero mean and variance 16. In Fig. 1 we plot the posterior probability density functions estimated by ETPS (top), ETKS (middle) with ensemble sizes $10^2$ (left), $10^3$ (center), and $10^4$ (right). The prior distribution is shown in red and the posterior estimated by IS with ensemble size $10^5$ is shown in black. We can see that ETPS provides with better approximation of the true probability density function. We

5 have also implemented ensemble Kalman Smoother with perturbed observations (SEnKS) developed by Burgers et al. (1998), that is used in an iterative ensemble Kalman smoother called multiple data assimilation of Emerick and Reynolds (2013). The posterior probability density function obtained by SEnKS is shown at the bottom of Fig. 1. The posterior is less skewed than the one provided by ETKS but otherwise is still not a good approximation of the true pdf. We will not use SEnKS for parameter estimation as it does not provide substantially different results than ETKS and both ETKS and ETPS are based on a

10 transformation, while SEnKS is not.



## 4 Single-phase Darcy flow

We consider a steady-state single-phase Darcy flow model defined over an aquifer of two-dimensional physical domain $D = [0,1] \times [0,1]$, which is given by,

$$-\nabla \cdot (k(x,y)\nabla P(x,y)) = f(x,y), \quad (x,y) \in D$$

$$P(x,y) = 0, \quad (x,y) \in \partial D$$

where $\nabla = (\partial/\partial x \ \partial/\partial y)^T$, $\cdot$ denotes the dot product, $P(x,y)$ the pressure, $k(x,y)$ the permeability, $f(x,y)$ the source term, which we assume to be $2\pi^2\cos(\pi x)\cos(\pi y)$, and $\partial D$ the boundary of domain $D$. The forward problem of this second order elliptical equation is to find the solution of pressure $P(x,y)$ for given $f(x,y)$ and $k(x,y)$. We, however, are interested in finding permeability given noisy observations of pressure at a few locations.

We perform numerical experiments with synthetic observations, where instead of a measuring device a model is used to obtain observations. We implement a cell-centered finite difference method to discretize the domain $D$ into $n \times n$ grid cells $X_i$ of size $\Delta x^2$ and solve the forward model with the true parameters. Then the synthetic observations are obtained by

$$\boldsymbol{y}_{\text{obs}} = \mathbf{L}(\mathbf{P}) + \eta,$$

where an element of $\mathbf{L}(\mathbf{P})$ is a linear functional of pressure, namely

$$L_l(\mathbf{P}) = \frac{1}{2\pi\sigma^2} \sum_{i=1}^{n^2} \exp\left(-\frac{||X_i - r_l||^2}{2\sigma^2}\right) P_i \Delta x^2, \quad l \in 1, \ldots, N_y$$

where $n = 50$, $\sigma = 0.01$ and $N_y = 16$, which is the number of observations. The observation locations are spread uniformly across the domain $D$ and $\eta$ denotes the observation noise drawn from a normal distribution with zero mean and standard deviation of 0.09.

### 4.1 Five parameter nonlinear problem

For our first numerical experiment with Darcy flow, we consider a low-dimensional problem where the permeability field is defined by mere 5 parameters similarly to Iglesias et al. (2014). We assume that the entire domain $D = [0,1] \times [0,1]$ is divided into two subdomains $D_1$ and $D_2$ as shown in Fig. 2. Each subdomain of $D$ represents a layer and is assumed to have a permeability function $k(\mathbf{X})$, where an element of $\mathbf{X}$ is defined by $X_i$ for $i = 1, \ldots, n^2$. The thickness of a layer on both sides $a$ and $b$, correspondingly, defines the slope of the interface and a parameter $c$ defines a vertical fault. The layer moves up or down

depending on $c < 0$ or $c > 0$, respectively, and its location is assumed to be fixed at $x = 0.5$.

Further, for this test case we assume piecewise constant permeability within each of the subdomains, hence $k(\mathbf{X})$ is given by

$$k(\mathbf{X}) = k_1\delta_{D_1}(\mathbf{X}) + k_2\delta_{D_2}(\mathbf{X}),$$

where $k_1$ and $k_2$ represent permeability of the subdomain $D_1$ and $D_2$, respectively, and $\delta$ is Dirac function. Then the parameters defining the permeability field for this configuration are

$$\boldsymbol{u} = (a \ b \ c \ \log(k_1) \ \log(k_2))^T.$$





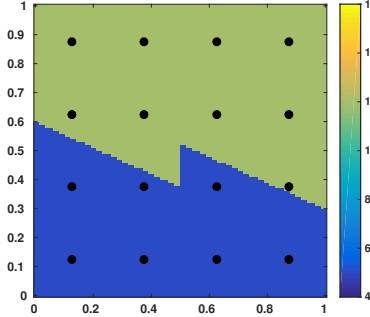

**Figure 2.** True permeability of the 5 parameter nonlinear problem with dots representing the observation locations.

We assume that the true parameters are $a^{\text{true}} = 0.6$, $b^{\text{true}} = 0.3$, $c^{\text{true}} = -0.15$, $k_1^{\text{true}} = 12$ and $k_2^{\text{true}} = 5$. These parameters are used to create synthetic observations. Figure 2 shows the true permeability with dots representing the observation locations. Next, we assume that the five uncertain parameters are drawn from a uniform distribution over a specified interval, namely $a, b \sim \mathcal{U}[0, 1]$, $c \sim \mathcal{U}[-0.5, 0.5]$, $k_1 \sim \mathcal{U}[10, 15]$ and $k_2 \sim \mathcal{U}[4, 7]$.

As it was pointed out in Sect. 2.2, ETPS updates the parameters within the original range of an initial ensemble, while ETKS does not. Therefore a change of variables has to be performed for ETKS so that the updated parameters are physically viable. In order to be consistent we perform the change of variables for ETPS as well. As the domain $D$ is $[0, 1] \times [0, 1]$, the parameters $a$ and $b$ should lie within the interval $[0, 1]$. To enforce this constraint we substitute $a$ according to

$$a' = \log\left(\frac{a}{1 - a}\right), \quad a' \in R$$

and similarly $b$ is substituted by $b'$. Thus the uncertain parameters are now $\boldsymbol{u}' = (a'\ b'\ c\ \log(k_1)\ \log(k_2))^T$.

In Fig. 3 we plot probability density functions for parameters $a$ (a)–(d), $c$ (e)–(h) and $\log(k_2)$ (i)–(l), as the parameters $b$ and $\log(k_1)$ show similar results. The posterior obtained by IS with ensemble size $10^6$ is plotted as a black line and the true value of parameters is plotted as a black line with crosses. The posterior of ETPS is shown at the top and the posterior of ETKS at the bottom. ETPS and ETKS used $10^3$ (odd columns) and $10^4$ (even columns) ensemble members. It is interesting to note that

ETKS overestimates the tails of the pdfs while ETPS underestimates them. While for the parameter $c$ pdf shown in Fig. 3(h) this is an advantage of ETKS, for the parameter $\log(k_2)$ pdf shown in Fig. 3(l) it is most certainly a disadvantage. ETKS optimises for the mean (and variance), which is better approximated by ETKS than by ETPS, as seen in Fig. 4(e). However this comes at a price of incorrect posterior shown in Fig. 3(k–l).

In order to avoid any bias due to an initial ensemble we perform 10 simulations based on a random draw of an initial

ensemble from the same prior distributions. We conduct the numerical experiments for ensemble sizes varying from 10 to $10^3$ with an increment of 50. In figure 4 we plot the true parameters $\boldsymbol{u}^{\text{true}}$, the mean $\bar{\bar{\boldsymbol{u}}}^a$ and the spread $\bar{\bar{\boldsymbol{u}}}^a \pm \bar{\boldsymbol{u}}_{\text{std}}^a$ of estimated




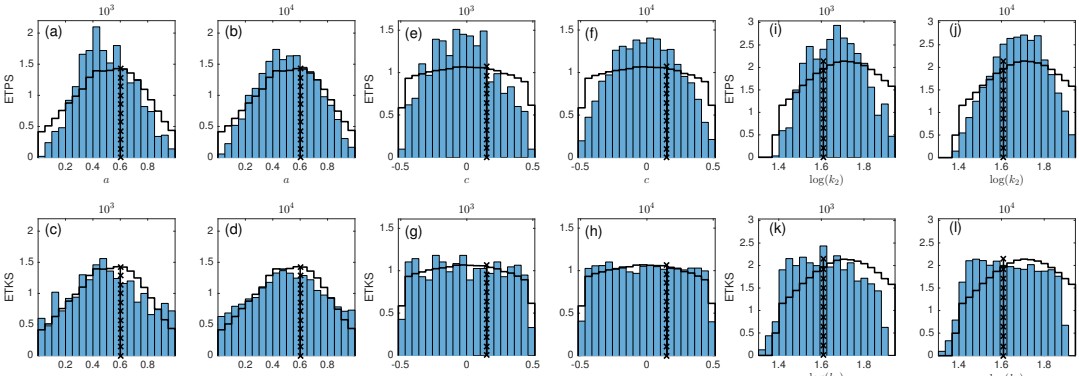

**Figure 3.** Probability density functions for the parameters $a$ (a)–(d), $c$ (e)–(h), and $\log(k_2)$ (i)–(l). The posterior obtained by IS with ensemble size $10^6$ is plotted as a black line and the true values of parameters are plotted as black crosses. The posterior of ETPS is shown at the top and the posterior of ETKS at the bottom. ETPS and ETKS used $10^3$ (odd columns) and $10^4$ (even columns) ensemble members.

parameters averaged over 10 simulations

$$\bar{\bar{u}}_i^a = \frac{1}{10}\sum_{r=1}^{10} \bar{u}_i^{a,r}, \quad \bar{u}_{\text{std}}^a = \frac{1}{10}\sum_{r=1}^{10}\sqrt{\frac{1}{M-1}\sum_{m=1}^{M}(u_{i,m}^{a,r}-\bar{u}_i^{a,r})^2}, \text{ where } \bar{u}_i^{a,r} = \frac{1}{M}\sum_{m=1}^{M} u_{i,m}^{a,r}, \ r=1,\ldots,10,$$

$M$ is ensemble size, $i=1,\ldots,5$ is parameter index, and the superscript $a$ is for the analysis. We observe that both data assimilation methods perform comparably in terms of mean estimation. The spread from ETPS is however smaller than from
ETKS for each parameter.

We compute an average of the relative error over all parameters

$$\text{RE}^{a,r} = \frac{1}{5}\sum_{i=1}^{5} \frac{|\bar{u}_i^{a,r} - u_i^{true}|}{|u_i^{true}|}, \ r=1,\ldots,10,$$

and the data misfit

$$\text{misfit}^{a,r} = (\bar{\boldsymbol{y}}^{a,r} - \boldsymbol{y}_{\text{obs}})^T R^{-1}(\bar{\boldsymbol{y}}^{a,r} - \boldsymbol{y}_{\text{obs}}), \ r=1,\ldots,10 \tag{13}$$

after data assimilation. The same metrics are computed before data assimilation and denoted by a superscript $b$. In Fig. 5(a)–(b) we plot $\left(\text{misfit}^{a,r} - \text{misfit}^{b,r}\right)$ and $\left(\text{RE}^{a,r} - \text{RE}^{b,r}\right)$, respectively, for each simulation $r$ as a function of ensemble size. ETPS is shown in blue and ETKS in red. Black line is at zero level. Positive values of the differences mean an increase of either data mismatch or relative error after data assimilation. We observe a data misfit decrease for both ETPS and ETKS except at an ensemble size 10. RE does not always decrease for ETPS: for some simulations ETPS is at zero level or slightly above it, while
for ETKS the sole exception is at an ensemble size 10.





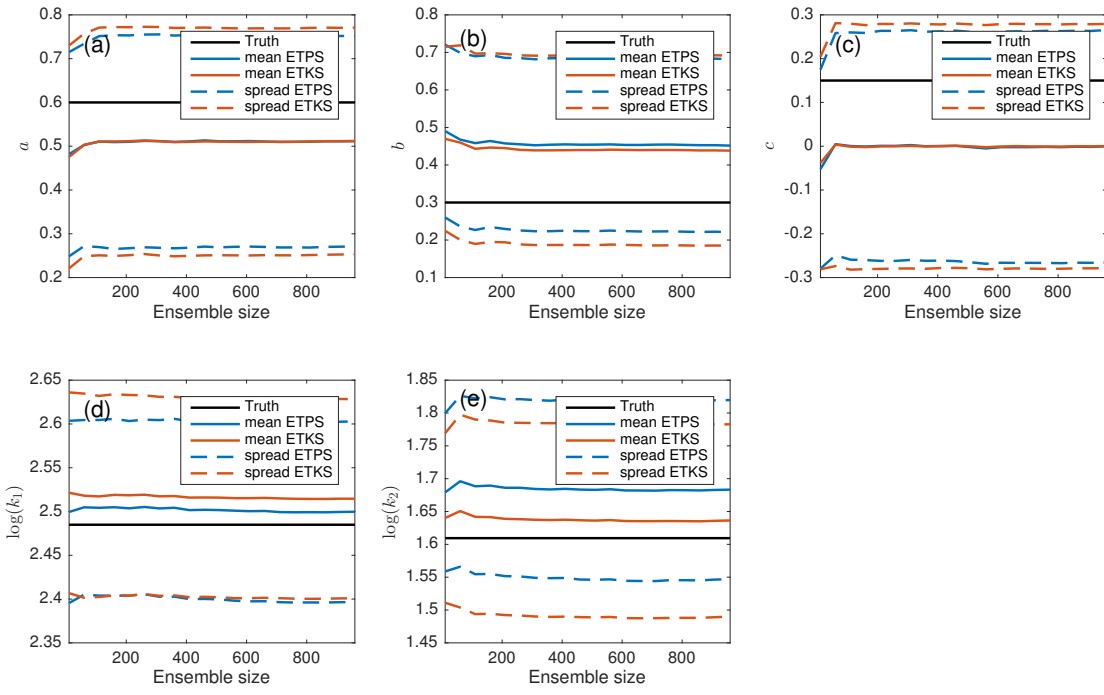

**Figure 4.** $\bar{\bar{\boldsymbol{u}}}^a$ and $\bar{\bar{\boldsymbol{u}}}^a \pm \bar{\boldsymbol{u}}^a_{\mathrm{std}}$ w.r.t ensemble size: (a) for the parameter $a$, (b) for $b$, (c) for $c$, (d) for $\log(k_1)$, (e) for $\log(k_2)$. ETPS is shown in blue, ETKS in red and the true parameters are in black.

## 4.2 High-dimensional nonlinear problem

Next, we consider a high-dimensional problem where the dimension of the uncertain parameter is $n^2 = 2500$. The domain $D$ is now not divided into subdomains. However, unlike in the previous test case here we implement a spatially varying permeability field. We assume the log permeability is generated by a random draw from a Gaussian distribution $\mathcal{N}(\log(\mathbf{5}), \mathbf{C})$. Here $\mathbf{5}$ is an

5  $n^2$ vector with all 5. $\mathbf{C}$ is assumed to be an exponential correlation with maximum correlation along $3\pi/4$, an element of $\mathbf{C}$ is

$$C_{i,j} = \exp(-3(|h_{i,j}|/v)), \; i,j = 1,\ldots,n^2.$$

Here $h_{i,j}$ is the distance between two spatial locations and $v$ is the correlation range which is taken to be $0.5$. As the covariance matrix is symmetric, we factorize it in upper and lower triangular matrices using Cholesky decomposition and denote the upper triangular matrix by $\mathbf{G}$. Next, we generate a vector $\mathcal{Z}$ of dimension $n^2$ from a Gaussian distribution with zero mean and

10  variance one, and create a permeability field $\log(\mathbf{k}(\mathbf{X}))$ across the domain $D$ according to Oliver et al. (2008):

$$\log(\mathbf{k}) = \log(\mathbf{5}) + \mathbf{G}^T \mathcal{Z},$$

The uncertain parameter is $\boldsymbol{u} = \mathcal{Z}^T$. Thus the dimension of the uncertain parameter is $n^2 = 2500$.





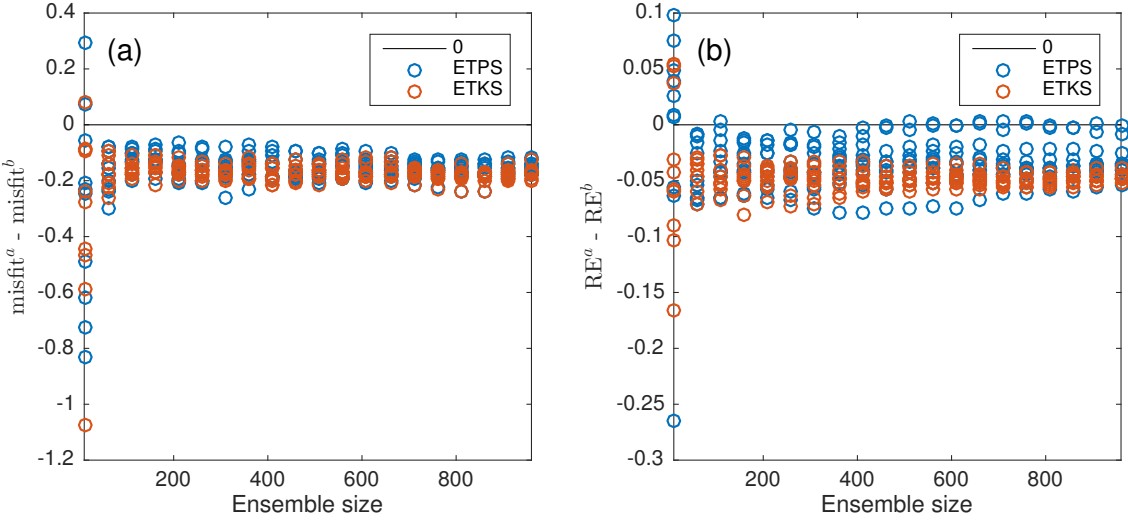

**Figure 5.** misfit$^{a,r}$ − misfit$^{b,r}$ (a) and RE$^{a,r}$ − RE$^{b,r}$ (b) w.r.t ensemble size. ETPS is shown in blue, ETKS in red and the zero level in black. A circle is for one simulation.

In Fig. 6 we plot the posterior pdf of first three modes $\mathcal{Z}_1$ (left), $\mathcal{Z}_2$ (center), and $\mathcal{Z}_3$ (right) obtained by IS with ensemble size $10^6$ and by ETPS (top) and ETKS (bottom) with ensemble size $10^4$. The posterior of these modes is roughly approximated by ETPS as shown in Fig. 6(a)–(c). This means that an ensemble size $10^4$ is not sufficient for ETPS to correctly estimate the posterior, when the total number of modes is large (2500). ETKS provides a skewed posterior of the modes shown in

5    Fig. 6(d)–(f), which was also observed in the one parameter nonlinear problem, see Fig. 1(f).

We perform 10 different simulations based on a random draw of an initial ensemble from the prior distribution. We conduct the numerical experiments for ensemble sizes varying from 10 to $10^3$ with an increment of 50. In Fig. 4 we plot mean and spread for $\mathcal{Z}_1$ (a), $\mathcal{Z}_2$ (b), and $\mathcal{Z}_3$ (c) averaged over 10 simulations in blue for ETPS and in red for ETKS. The true variables are shown in black. We observe that in terms of the mean estimation of the first three modes ETPS outperforms ETKS. The

10    spread provided by ETPS is smaller than the one provided by ETKS, as in the previous test cases.

We compute the root mean square error (RMSE) of the log permeability field

$$\text{RMSE}^{r,a} = \sqrt{\frac{1}{n^2}(\overline{\mathcal{Z}}^{a,r} - \mathcal{Z}^{\text{true}})^T \mathbf{C}(\overline{\mathcal{Z}}^{a,r} - \mathcal{Z}^{\text{true}})}, \quad r = 1, \dots, 10,$$

and variance

$$\text{variance}^{r,a} = \frac{1}{M-1} \sum_{m=1}^{M} (\mathcal{Z}_m^{a,r} - \overline{\mathcal{Z}}^{a,r})^T \mathbf{C}(\mathcal{Z}_m^{a,r} - \overline{\mathcal{Z}}^{a,r}), \quad r = 1, \dots, 10.$$



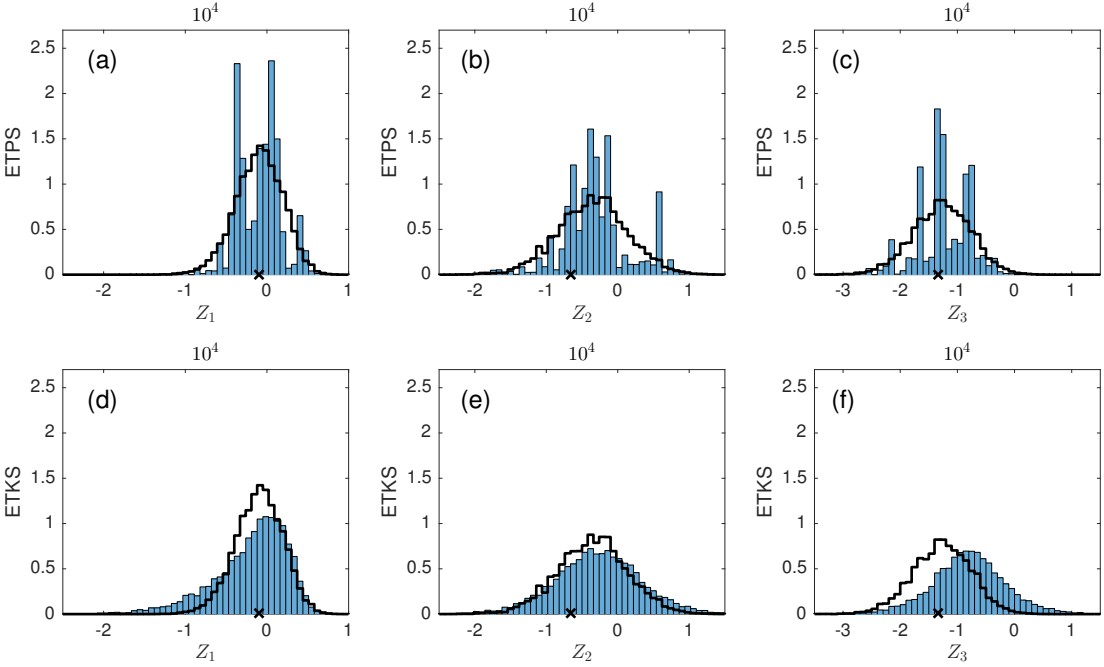

**Figure 6.** The posterior probability density function of parameters $\mathcal{Z}_1$ (left), $\mathcal{Z}_2$ (center), and $\mathcal{Z}_3$ (right). The posterior obtained by IS with ensemble size $10^6$ is plotted as a black line and the true parameter as a black cross. The posterior of ETPS is shown at the top and the posterior of ETKS at the bottom. Both ETPS and ETKS used $10^4$ ensemble members.

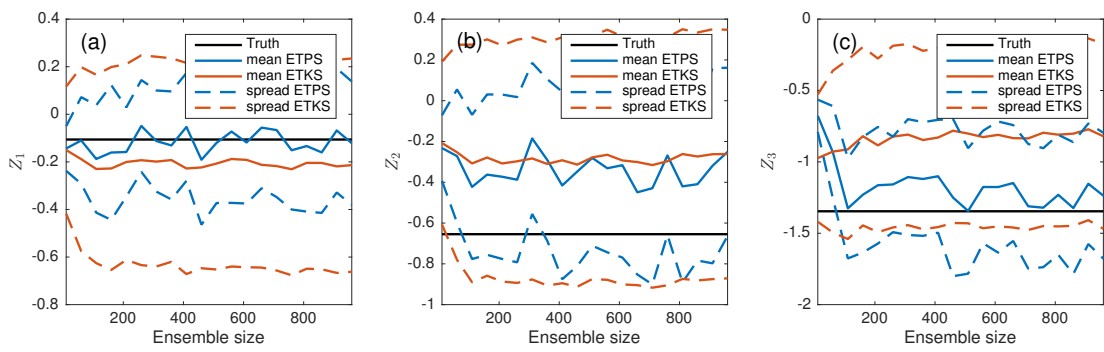

**Figure 7.** Mean and spread for $\mathcal{Z}_1$ (a), $\mathcal{Z}_2$ (b), and $\mathcal{Z}_3$ (c) w.r.t ensemble size. ETPS is shown in blue, ETKS in red and the true parameters are in black.




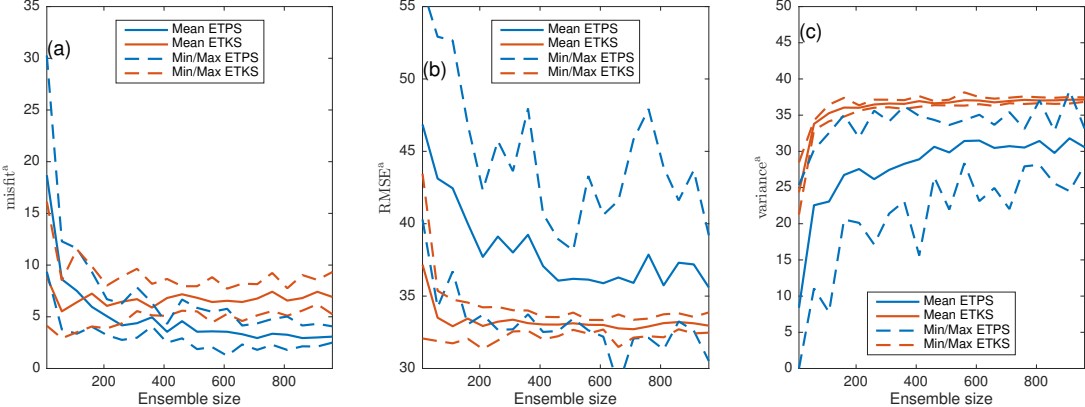

**Figure 8.** Mean, minimum and maximum over 10 simulations after data assimilation for the data misfit (a), RMSE (b), and variance (c). ETPS is shown in blue and ETKS in red.

We also compute the data misfit for each simulation after data assimilation by Eq. (13). In Fig. 8 we plot mean, minimum and maximum over 10 simulations after data assimilation for the data misfit (left), RMSE (center), and variance (right). ETPS is shown in blue and ETKS in red. We observe that ETPS in underdispersive compared to ETKS. This is due to the linear transformation, as it results in some ensemble members being nearly identical. Therefore a perturbation of ensemble members

is needed, which could be performed based on random walk. This is, however, out of the scope of this paper. Misfit given by ETPS is smaller than the one given by ETKS for almost all simulations at ensemble sizes greater than 150. The RMSE on the contrary is larger. In Fig. 9(a)–(b) we plot $(\text{misfit}^{a,r} - \text{misfit}^{b,r})$ and $(\text{RE}^{a,r} - \text{RE}^{b,r})$, respectively, as a function of ensemble size for a simulation $r = 1, \ldots, 10$. The superscript $b$ is for the metrics before data assimilation and the superscript $a$ is for the metrics after data assimilation. ETKS always provides a decrease in both the data misfit and RMSE except at ensemble size 10.

ETPS gives a decrease in the data misfit though an increase in RMSE. However, as the ensemble size increases this happens less often as can be seen in Fig. 9(c), where we plot for ETPS a percentage of simulations that result in $(\text{RMSE}^a - \text{RMSE}^b) > 0$ and a linear fit as a function of ensemble size.

In Fig. 10 we plot log permeability fields. In Fig. 10(a) the true permeability is shown with dots representing the observation locations, and in Fig. 10(d) the mean permeability field obtained by IS with ensemble size $10^5$. The RMSE provided by IS

is 32.62. In Fig. 10(b–e) and Fig. 10(c–f) we display mean permeability fields obtained with ensemble size $10^3$ by ETPS and ETKS, respectively. In Fig. 10(b–c) we plot the mean log permeabilities for the smallest RMSE over simulations, which is 30.51 for ETPS and 32.48 for ETKS. In Fig. 10(d–e) we plot the mean log permeabilities for the largest RMSE over simulations, which is 39.2 for ETPS and 33.9 for ETKS. We observe that ETKS as well as IS provide smooth mean permeability fields that have smaller absolute values than the true permeability. ETPS gives higher variations of the mean permeability field and in an

excellent agreement with the true permeability for a good initial ensemble shown in Fig. 10(b). However, it remains unclear how to choose a good initial ensemble. It should be noted that IS with ensemble size $10^3$ and this good initial ensemble gives





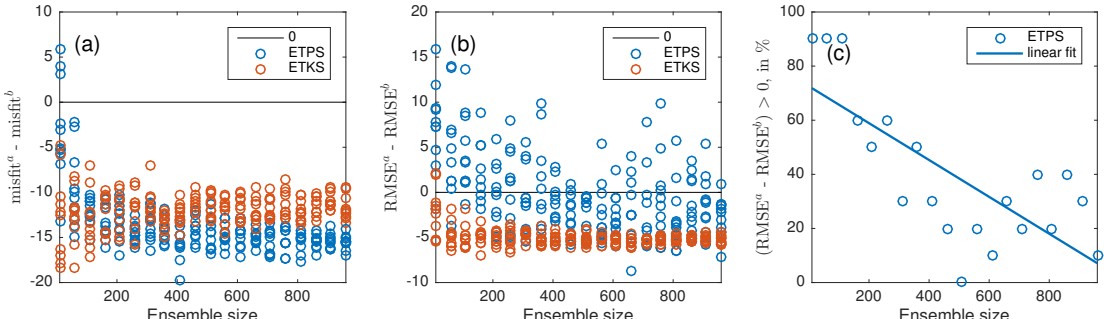

**Figure 9.** misfit$^{a,r}$ − misfit$^{b,r}$ (a) and RE$^{a,r}$ − RE$^{b,r}$ (b) w.r.t ensemble size. ETPS is shown in blue, ETKS in red and zero level in black. One circle is for one simulation. For ETPS % of simulations that result in (RMSE$^a$ − RMSE$^b$) > 0 and a linear fit as a function of ensemble size are shown in (c).

**Table 1.** Optimal localization radius for ETPS and ETKS at different ensemble sizes M.

| M | 10 | 110 | 210 | ... | 910 |
|------|-----|-----|-----|-----|-----|
| ETPS | 0.2 | 0.6 | 0.8 | ... | 0.8 |
| ETKS | 0.2 | 1.2 | 1.2 | ... | 1.2 |

the RMSE 30.51 and the same mean log permeability field as ETPS shown in Fig. 10(b). However, IS does not change the parameters, only their weights, while ETPS does change the parameters. Therefore ETPS has an advantage of IS representing the correct posterior but does not have its disadvantage of resampling lacking.

In Fig. 11 we plot variance of the permeability fields obtained with ensemble size $10^5$ by IS (d), with ensemble size $10^3$
by ETPS (b–e) and ETKS (c–f) . ETPS (b) and ETKS (c) are for the smallest RMSE, and ETPS (e) and ETKS (f) are for the largest RMSE over simulations. ETKS again provides smoother variance than ETPS.

Next we employ localization for both ETPS and ETKS. Optimal localization radius in terms of the smallest RMSE was obtained for one simulation (shown in Table 1) and fixed for the remaining 9 simulations. In Fig. 12(a–c) we plot change in misfit, RMSE and the percentage of simulations for which RMSE of ETPS increased after data assimilation, respectively. ETKS
with localization gives an equivalent performance as without localization at large ensemble sizes, which is to be expected as the optimal localization radius is large. For the small ensemble size $M = 10$ ETKS with localization performs worse than without localization. This could be related to a need of more advanced losalization methods based on wavelets rather than on distance, see Chen and Oliver (2012). ETPS on the other hand highly improved when localization was applied: for ensemble sizes greater than 300 all simulations result in the RMSE decrease after data assimilation. The RMSE decreased and the variance increased
for all ensemble sizes as shown in Fig. 13(b–c) compared to Fig. 8(b–c). The posterior estimations, however, degraded: the pdf shown in Fig. 14(a–c) with ensemble size $10^4$ resembles the posterior obtained by ETKS shown in Fig. 6(d–e). Therefore on the one hand localization improved the RMSE behaviour for all simulations but on the other hand it deteriorated the posterior



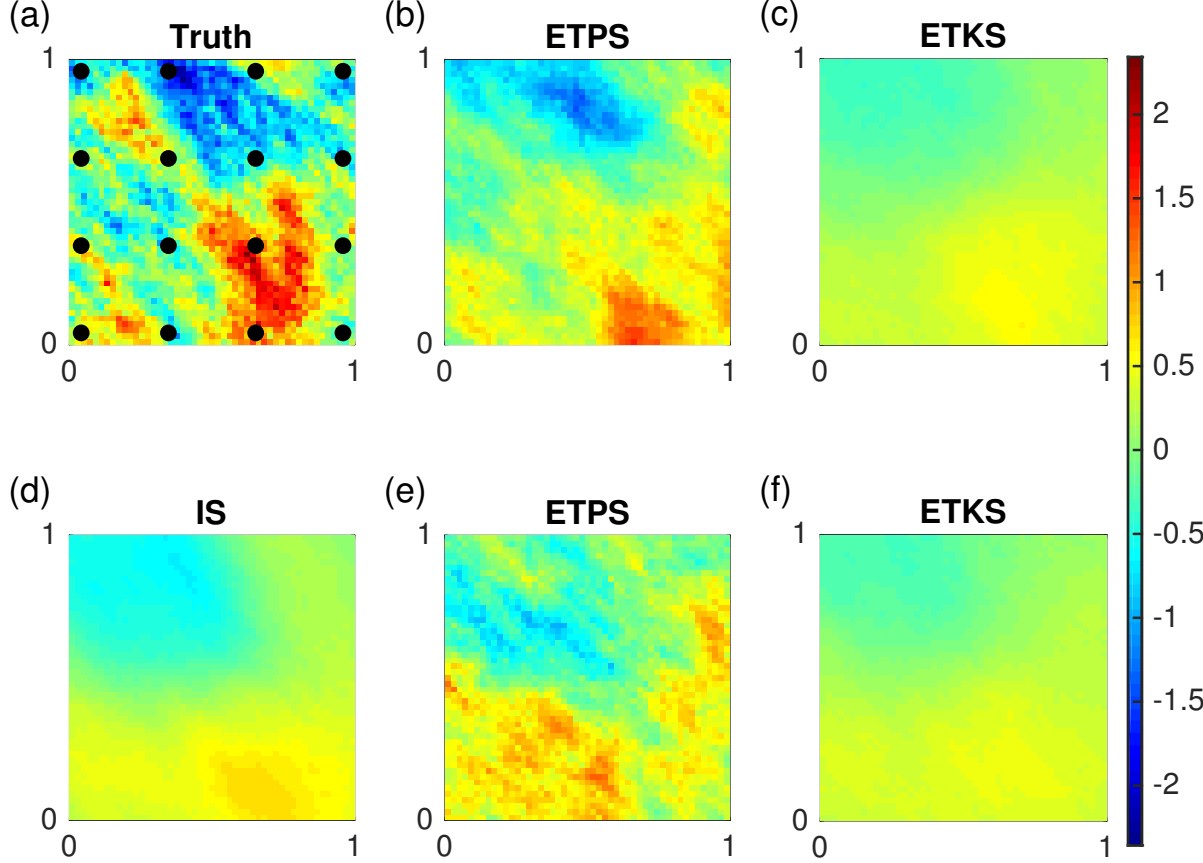

**Figure 10.** Log permeability field: truth with dots representing the observation locations (a), mean obtained with ensemble size $10^5$ by IS (d), mean obtained with ensemble size $10^3$ by ETPS (b–e) and by ETKS (c–f). Mean at the smallest RMSE (b–c) and at the largest RMSE (e–f) over simulations.

estimation. The increase in RMSE after data assimilation without localization could be related to a substantial adjustment of the uncertain parameters. A possible solution to this problem is an iterative approach to data assimilation as it has been done for ensemble Kalman smoothers (e.g Chen and Oliver, 2013; Emerick and Reynolds, 2013; Bocquet and Sakov, 2014) and will be a subject of our future study.



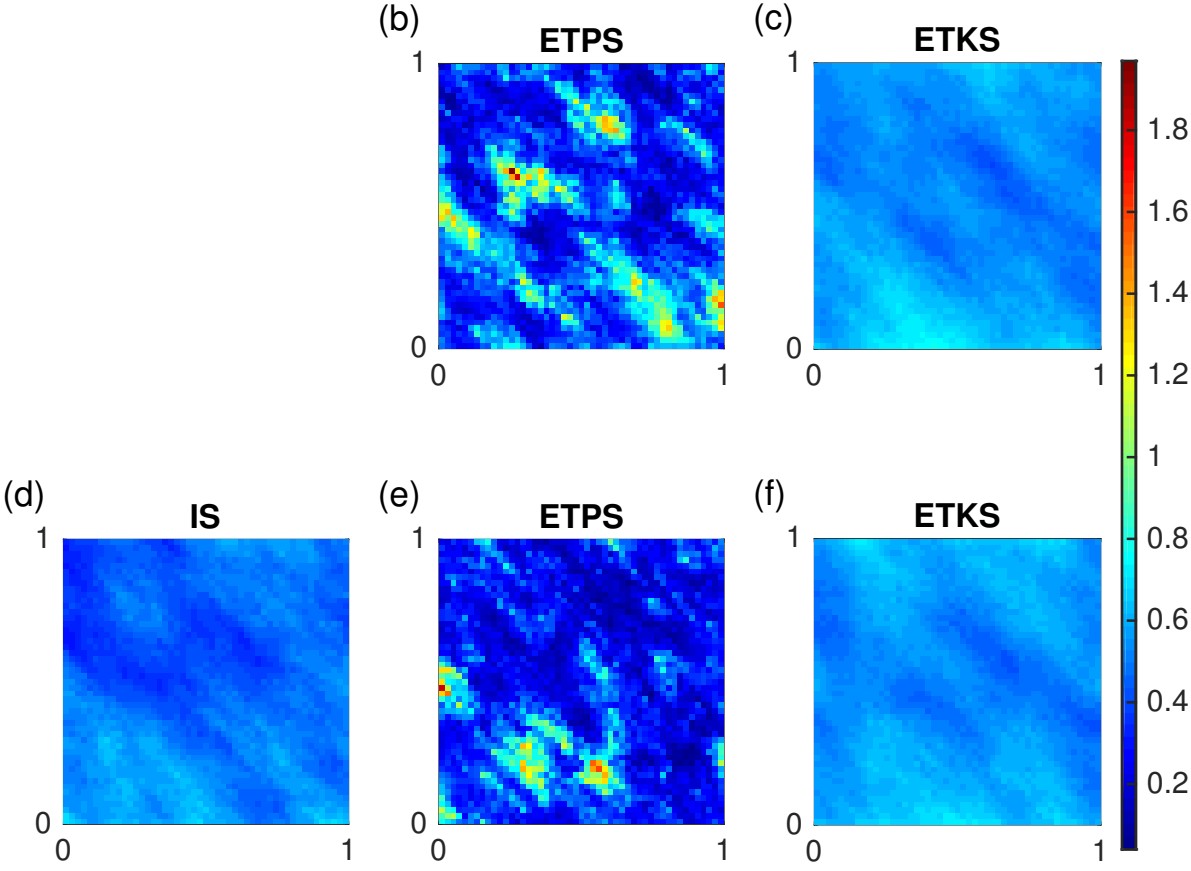

**Figure 11.** Variance of log permeability fields: obtained with ensemble size $10^5$ by IS (d), with ensemble size $10^3$ by ETPS (b–e), and ETKS (c–f). Variance at the smallest RMSE (b–c) and at the largest RMSE (e–f) over simulations.

## 5  Conclusions

MCMC methods remain the most reliable methods for estimating the posterior distributions of uncertain model parameters and states. They, however, also remain computationally expensive. Ensemble Kalman filters (or smoothers) provide computationally affordable approximations but rely on the assumptions of Gaussian probabilities. For nonlinear models even if the prior is

5    Gaussian the posterior is not Gaussian anymore. Particle filtering (or importance sampling) on the other hand does not have such an assumption but requires a resampling step, which is a challenge for parameter estimation. Ensemble transform particle filter (or smoother) is a particle filtering method that resamples the particles (samples) based on their importance weights and



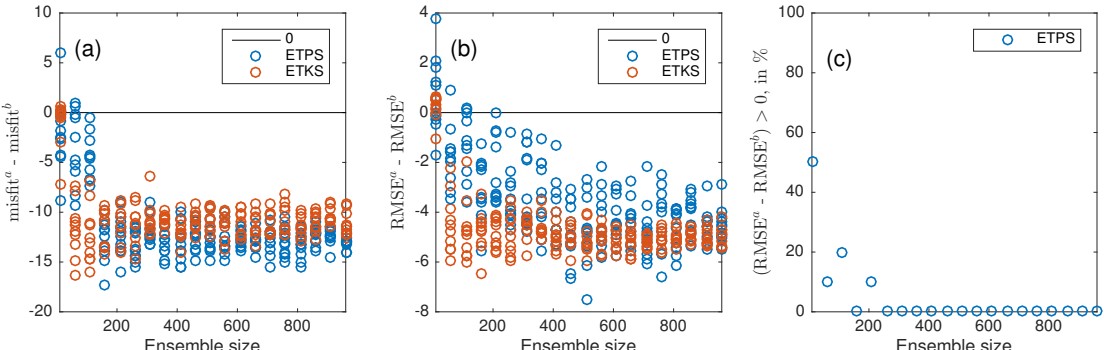

**Figure 12.** Same as figure 9, but with localization.

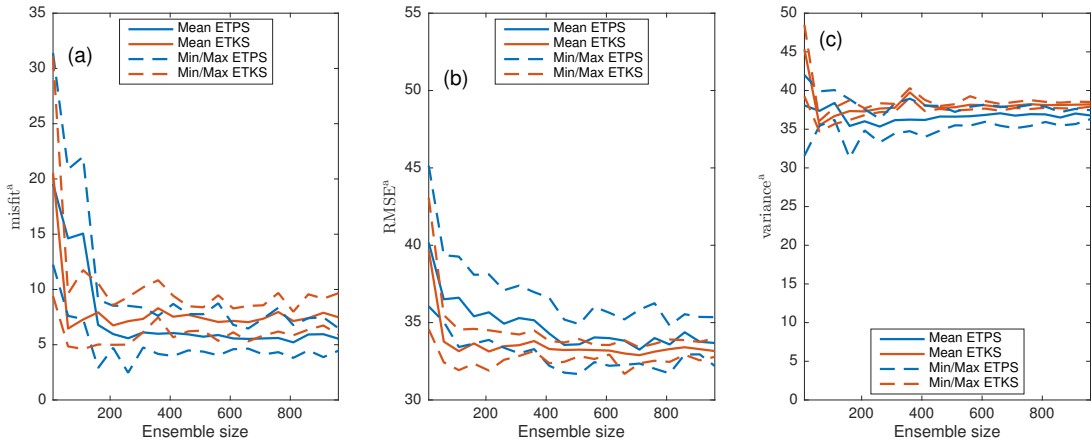

**Figure 13.** Same as figure 8, but with localization.

covariance maximization among the particles. Therefore it has an importance sampling advantage of predicting the correct posterior but does not have its disadvantage of resampling lacking.

In this paper, we have shown that ETPS outperforms ETKS for the posterior estimations in both low- and high-dimensional nonlinear problems. Moreover as the ensemble size increases the posterior of ETPS converges to the posterior of IS with a
5   large ensemble, while the posterior of ETKS remains unchanged. However, in the high-dimensional problem of 2500 uncertain parameters for some simulations ETPS gives an increase in the RMSE after data assimilation. This issue is resolved once distance-based localization is applied, which however deteriorated the posterior estimation.

*Competing interests.* The authors declare that they have no conflict of interest.



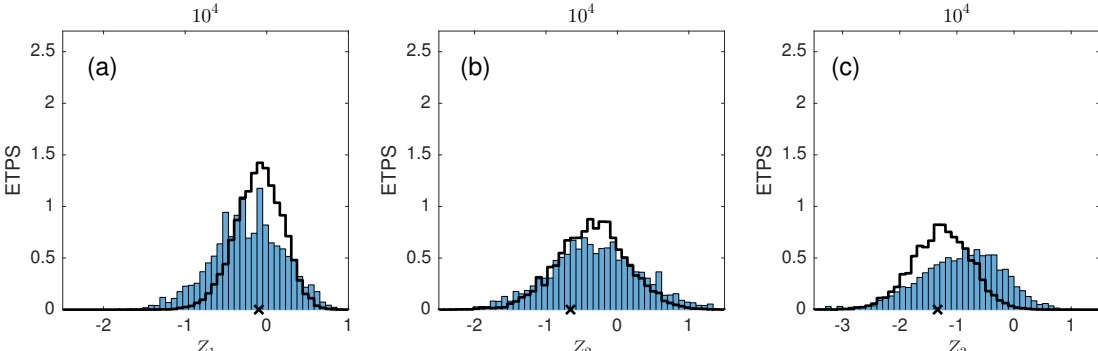

**Figure 14.** The posterior probability density function of parameters $\mathcal{Z}_1$ (a), $\mathcal{Z}_2$ (b), and $\mathcal{Z}_3$ (c). The posterior obtained by IS with ensemble size $10^6$ is plotted as a black line and the true parameter as a black cross. The posterior of ETPS with localization used $10^4$ ensemble members.

*Acknowledgements.* This work is part of the research programme Shell-NWO/FOM Computational Sciences for Energy Research (CSER) with project number 14CSER007 which is partly financed by the Netherlands Organization for Scientific Research (NWO).



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
