# Peer review of "Application of ensemble transform data assimilation methods for parameter estimation in reservoir modelling"

_Nonlinear Processes in Geophysics, 2018_

## Referee Comment (RC1) · Anonymous Referee #1 · 20 Apr 2018

[Comments to the Authors]

This manuscript aims to compare performance of ensemble transform Kalman smoother (ETKS) and ensemble transform particle smoother (ETPS) in nonlinear parameter estimation problem. The authors conducted observing system simulation experiments and obtained reasonable results. The scope discussed in this manuscript suits well to Nonlinear Processes in Geophysics. I do not have any major concerns for the experiments presented in the manuscript. However, some discussions and descriptions are difficult to follow due to insufficient explanation. Here I list the concerns, which would be beneficial to improve the manuscript further.

[Figure]

[General Comments]

1. Scientific Significance: The authors addressed that they applied the ETPS for estimating a large number of uncertain parameters (P2L34). It can be a good motivation; however I could not understand the scientific significance that can be achieved by applying the ETPS and ETKS for the large-dimensional problem. Please address this point clearly in abstract and conclusion.

2. Lack of explanations: I could not follow several logics of the manuscript, therefore, my major comments includes many "whys" and "reasons". Most of the issues should be solved by adding sufficient explanations.

3. Results (Figures): Some figures were discussed insufficiently. It is better to remove figure(s) if they are not needed.

4. Methods: The author compared the ETKS and ETPS. I am wondering the difference between the ETPS used in this study and a nonlinear ensemble transform filter by Tödter and Ahrens (2015). Also, it is better to compare the localization methodology with local particle filters (Penny and Miyoshi 2016; Poterjoy 2016). Please add more discussion on difference from existing methods.

Penny, S. G. and T. Miyoshi, 2016: A local particle filter for high-dimensional geophysical systems. Nonlin. Processes Geophys., 23, 391-405.

Poterjoy, J. (2016). A localized particle filter for high-dimensional nonlinear systems. Monthly Weather Review, 144(1), 59-76.

Tödter, J., and B. Ahrens, 2015: A second-order exact ensemble square root filter for nonlinear data assimilation. Mon. Wea. Rev., 143, 1347–1367.

[Major Comments]

1. P1L13: Please add reason(s) why ETPS is very sensitive w.r.t. the initial ensemble.

2. P1L15: Please add reason(s) why the localization deteriorated the posterior estimation

3. P7L15: Isn't it possible to apply the localization between variables?

4. P8L1: I could not understand the sentence "$\sim$ is made such that y_obs=48". Please rephrase this sentence.

5. P10L14: Please explain more about reason(s).

6. P11L5: Why? Does it relate to the resampling issue discussed later?

7. Fig.4, Fig. 8 (b) and (c), : I did not understand why this figure is needed because they were not discussed.

8. P15L5, perturbation of ensemble member: In generic PF, the resampling (or inflation) method is very important to avoid the particle convergence. Could you explain why you did not need to consider this issue?

9. Fig. 8 (b): I was confused why the ETKS outperforms the ETPS if RMSE is used for the metric.

10. Fig. 10: It is helpful to add RMSEs on the figure.

11. Table 1: Could you discuss why the optimal radius for the ETKS is larger than that of the ETPS?

12. P16L15: Please discuss why the localization degrades the posterior estimation

13. Conclusion: It would be helpful to add findings and limitation further in this section.

---

## Referee Comment (RC2) · Anonymous Referee #2 · 30 Apr 2018

This article presents a comparison between an EnKF and a particle filter based approaches for parameter estimation in a time independent model. I think that this comparison is relevant and can provide good insights about the performance of these two approaches in the context of parameter estimation. However I found many aspects that needs further clarification to support the conclusions made by the authors of this work. Major revisions are required to the paper.

Major points -It is not clear if Importance Sampling and particle filters can be treated as synonyms. From my point of view particle filters can include different approaches for particle resampling to avoid the collapse of the filter and this is different from Impor-

tance Sampling which in principle does not include the resampling step.

-Page 2, 30 it is stated that particle filters do not update the uncertain parameters. This is not correct, many particle filters with different resampling approaches has been developed. These resampling steps introduce changes in the uncertain parameters so they get closer to the ones that produce the maximum observation likelihood, so the parameter ensemble evolves with time. It is true that the proposed technique performs this in a different way introducing a deterministic update of the parameter values (while usually resampling techniques in particle filters are stochastic). The difference between the implemented technique and previous techniques should be more clearly stated. - In this work the implemented techniques are described as smoothers, however all the experiments performed are time independent. It is not clear for me what would be the difference between a filter or a smoother if there is no time involved. Please clarify this point. In the methodology I cannot find a difference between the filter implementation or the smoother implementation since there are no time index in the equations. -Page 3, near 5: it is stated that ETKS does not employ the correlation in the estimation of the parameter. Filter equations are solved in the space defined by the ensemble members, but this implementation is basically equivalent to other EnKF which relies on the correlation between uncertain parameters and observed variables. Please clarify this point. -Page 8, 5 an iterative Kalman Smoother is mentioned here and shown in Figure 1, but detailed information about this technique is lacking. I suggest removing this technique since it has not been used in the experiments with the Darcy flow and also it has not been described in detail in the methodology section.

-In Figure 1, d, e and f a Gaussian prior produces a non-Gaussian posterior using ETKS. Since the EnKF relies on the linear and Gaussian assumption is it possible to obtain a non-Gaussian posterior from a Gaussian prior? -What is the motivation behind the functional introduced to define the observations in page 9, 15? What is $r\_l$ which appears in the definition of $L\_l(P)$? -Figure 6 shows the distribution for the first 3 modes of Z. Please clarify how these modes are obtained. -Figure 8 shows that the RMSE

associated with ETKS is always lower than the RMSE for ETPS, however the first 3 moments of Z are better estimated by ETPS than for ETKS. Does this mean that ETKS provides a better estimation of higher order modes? -IS and ETKS provide spatially smother solutions than ETPS (Figure 10), however ETPS seems to provide a better representation of the spatial variability and patters of the parameter. The explanation provided by the authors is not convincing for me. IS with a large number of particles should provide a very good estimation of the parameters (this approach is used as a benchmark by the authors). Also the distribution for the first 3 moments of Z are relatively similar between ETPS, ETKS and IS (but the spatial variability shown in Figure 10 are very different). This point is very important and I think it should be explored and discussed in more detail. -The authors show that in many cases ETPS improves the fitting to the observations but degrades the RMSE of the parameter. Can this be due to an over fitting of the observations? -For the experiments including localization, the authors do not show the spatial distribution of the estimated parameters. This is very important since using localization can significantly improve the small scale details in the estimated parameter field. This figure should be included in order to better evaluate the impact of localization. It is also strange that there is almost no improvement between the global and local implementation of the ETKS algorithm. With such a large number of variables and for the smaller ensemble sizes a larger positive impact would be usually expected. -The degradation of the ETKS with a small ensemble size using localization is unexpected. The authors indicate that better localization approaches should be used but previous studies usually indicates that the impact of localization is stronger for smaller ensemble sizes. Are other works that shows this kind of behavior with localization degrading the performance of the filter for small ensemble sizes? -Page 16, before 5, it is stated that "However, IS does not change the parameters, only their weights, while ETPS does change the parameters. Therefore ETPS has an advantage of IS representing the correct posterior but does not have its disadvantage of resampling lacking". If the posterior is correct and tacking into account that there is no time evolution in this context, what would be the problem with the lacking of resampling

in the IS? The results described in this section also suggest that the solutions provided by IS and ETPS are very similar given that the initial condition is the same (once again resampling does not seems to be an issue in this context).

-Does ETPS with $10^5$ ensemble members produce a smooth field like the one produced by IS? In other words, the spatial variability that we see in Figure 10 b is produced by sampling errors or is the result of a better estimation of the parameter field? Results mentioned in the previous comment suggests that spatial variability is just a result of sampling noise and because of that is extremely sensitive to the prior ensemble. If we have a "lucky" prior then we end up with good results, but if the prior is bad then the result is also bad. In this sense ETKF seems to be more robust (which is reasonable when we need to update a large number of parameters with a relatively small ensemble and when the posterior distribution is not too far from a Gaussian). -Conclusions, page 19, 5: It is stated that ETPS better fit the posterior. However if we look at Figure 6 we found that for $10^4$ particles (which is a large ensemble for most applications), ETPS fit is very noisy. Can the authors perform and objective comparison between the posterior provided by IS and the posterior provided by ETPS and ETKS (for instance using the Kullback-Leibler divergence or other objective comparison between two distributions). -Conclusions: Conclusions are very optimistic with respect to the performance of ETPS, however the RMSE of ETKS is always better in the large parameter space experiments. This suggests that the mean of the posterior is better estimated by ETKS rather than ETPS. While the mean is usually used as the best estimator of the parameter value, this should be mentioned in the conclusions. Minor points Page 12, 5: It is stated that is assumed to be an exponential correlation with maximum correlation along 3pi/4 ... It is not clear for me the meaning of this sentence. Page 7, 20: It is stated that R0 approximation is used with large ensembles in the experiments presented in this work, but in the result section it is not clear if this approximation has been used or not. Figure 10, It would be nice to include grid lines or to include the observation location in all the panels just to have a reference to compare smaller scale details in the estimated parameters.

---

## Author Response (AR1)

Reply to Referee 1

This manuscript aims to compare performance of ensemble transform Kalman smoother (ETKS) and ensemble transform particle smoother (ETPS) in nonlinear parameter estimation problem. The authors conducted observing system simulation experiments and obtained reasonable results. The scope discussed in this manuscript suits well to Nonlinear Processes in Geophysics. I do not have any major concerns for the experiments presented in the manuscript. However, some discussions and descriptions are difficult to follow due to insufficient explanation. Here I list the concerns, which would be beneficial to improve the manuscript further.

[General Comments]
1. Scientific Significance: The authors addressed that they applied the ETPS for estimating a large number of uncertain parameters (P2L34). It can be a good motivation; however I could not understand the scientific significance that can be achieved by applying the ETPS and ETKS for the large-dimensional problem. Please address this point clearly in abstract and conclusion.

Reply: The large number of uncertain parameters is of a particular interest for subsurface reservoir modelling as it allows to parameterise permeability on the grid. The most reliable methods of MCMC are computationally expensive and sequential ensemble methods such as ensemble Kalman filers and particle filters provide with a favourable alternative.

2. Lack of explanations: I could not follow several logics of the manuscript, therefore, my major comments includes many "whys" and "reasons". Most of the issues should be solved by adding sufficient explanations.

Reply: we have added more explanations (please see point-by-point answer).

3. Results (Figures): Some figures were discussed insufficiently. It is better to remove figure(s) if they are not needed.

Reply: we have adopted the revised version accordingly.

4. Methods: The author compared the ETKS and ETPS. I am wondering the difference between the ETPS used in this study and a nonlinear ensemble transform filter by Tödter and Ahrens (2015) (Tödter, J., and B. Ahrens, 2015: A second-order exact ensemble square root filter for nonlinear data assimilation. Mon. Wea. Rev., 143, 1347–1367).

Reply: The paper by Tödter and Ahrens (2015) addresses an important issue of ensemble Kalman (square root) filter being biased for nonlinear models and makes a correction for that. The resulting algorithm does not attempt to estimate the full analysis pdf in contrast to the ETPF.

Also, it is better to compare the localization methodology with local particle filters (Penny and Miyoshi 2016; Poterjoy 2016). Please add more discussion on difference from existing methods.

Penny, S. G. and T. Miyoshi, 2016: A local particle filter for high-dimensional geophysical systems. Nonlin. Processes Geophys., 23, 391-405. Poterjoy, J. (2016). A localized particle filter for high-dimensional nonlinear systems. Monthly Weather Review, 144(1), 59-76.

Reply: The localization methodology considered in this manuscript was particularly developed for ETPF by S. Reich and C. Cotter (2015). The same holds for localization for ensemble transform Kalman filter by Hunt at al (2007). Therefore we keep this comparison. However, we mention Penny and Miyoshi 2016; Poterjoy 2016 in the localization section to give a flavour of an ongoing research in local particle filters.

[Major Comments]
1. P1L13: Please add reason(s) why ETPS is very sensitive w.r.t. the initial ensemble.

Reply: ETKF is very robust while ETPF is very sensitive with respect to the initial ensemble due to a sampling error.

2. P1L15: Please add reason(s) why the localization deteriorated the posterior estimation.

Reply: An issue of an increase in the root mean square error after data assimilation is performed in ETPF for a high-dimensional test problem is resolved by applying distance-based localization, which however deteriorated the posterior estimation of the first mode by largely increasing the variance, which is due to a combination of less varying localized weights, not keeping the imposed bounds on the modes via the Karhunen-Loeve expansion and the main variability explained by the first mode.

3. P7L15: Isn't it possible to apply the localization between variables?

Reply: Localization can be defined for variables that depend on space. Thus it cannot be applied to geometrical parameters, for example.

4. P8L1: I could not understand the sentence " is made such that $y_{\mathrm{obs}} = 48$". Please rephrase this sentence.

Reply: The true parameter $u^{\mathrm{true}}$ gives $h(u^{\mathrm{true}}) = 48$.

5. P10L14: Please explain more about reason(s).

Reply: It is interesting to note that ETKF overestimates the tails of the pdfs while ETPF underestimates them, which indicates that there is not enough spread in the ensemble.

6. P11L5: Why? Does it relate to the resampling issue discussed later?

Reply: ETPF provides ensemble members that stay within the original bounds, while ETKF—outside the bounds. Moreover the optimal transport problem solved by ETPF results in some particles being almost identical. Therefore ETPF gives smaller spread than ETKF.

7. Fig.4, Fig. 8 (b) and (c), : I did not understand why this figure is needed because they were not discussed.

Reply: In Fig. 4 we plot the true parameters $\mathbf{u}^{\text{true}}$, the mean $\bar{\bar{\mathbf{u}}}^a$ and the spread $\bar{\bar{\mathbf{u}}}^a \pm \bar{\mathbf{u}}^a_{\text{std}}$ of estimated parameters averaged over 10 simulations.
In Fig. 8 (in the revised version it is Fig. 6) we plot mean, minimum and maximum over 10 simulations after data assimilation for the data misfit (a), RMSE (b), and variance (c). ETPF is shown in blue and ETKF in red. We observe that ETPF in underdispersive compared to ETKF (c). This is due to the linear transformation, as it results in some ensemble members being nearly identical. Misfit (a) given by ETPF is smaller than the one given by ETKF for almost all simulations at ensemble sizes greater than 150. The RMSE(b) on the contrary is larger.

8. P15L5, perturbation of ensemble member: In generic PF, the resampling (or inflation) method is very important to avoid the particle convergence. Could you explain why you did not need to consider this issue?

Reply: The model is time independent thus the issue of collapse does not rise here.

9. Fig. 8 (b): I was confused why the ETKS outperforms the ETPS if RMSE is used for the metric.

Reply: The first 3 modes $\mathcal{Z}$ are better estimated by ETPF than by ETKF, thus the permeability field defined only by those modes gives better resemblance to the true permeability when approximated by ETPF (please see Fig.13 in the revised version). ETKF, however, provides a better estimation of higher order modes, thus ETKF outperforms the ETPF if RMSE is used for the metric taking into account all modes.

10. Fig. 10: It is helpful to add RMSEs on the figure.
Reply: Agreed, it is added.

11. Table 1: Could you discuss why the optimal radius for the ETKS is larger than that of the ETPS?

Reply: It was also observed by Cheng and Reich (2015) that the localization radius for ETKF is larger than for ETPF. This is probably related to more noisy approximation of the posterior by ETPF than by ETKF.
Y. Chen and S. Reich, Data assimilation: a dynamical system perspective. Frontiers in Applied Dynamical Systems: Reviews and Tutorials Vol 2, 75-118, 2015

12. P16L15: Please discuss why the localization degrades the posterior estimation.

Reply: The posterior estimation of the first mode $\mathcal{Z}_1$ degraded, while of others improved. The Kullback-Leibler divergence for the first mode is 0.73 (compare to 0.21 without localization), and for second and third is 0.2 and 0.18,

correspondently (compare to 0.42 and 0.6 without localization). Variance of the posteriors is larger when localization is applied for both ETPF and ETKF. The localized weights given by Eq. 11 vary less than the non-localized weights given by Eq. 3. Therefore the localized pdf is less noisy than the non-localized. However, localization applied in the form of the Karhunen-Loeve expansion given by Eq. 14 does not retain the imposed bounds on the modes $\mathcal{Z}$ as we need to invert a matrix product of eigenvalue and eigenvector matrices to obtain the modes. By increasing the localization radius to 1.2 we get the Kullback-Leibler divergence 0.65 for the first mode, and 0.14 and 0.12 for the second and third, correspondently, thus the posterior approximation improved only slightly.

13. Conclusion: It would be helpful to add findings and limitation further in this section.

Reply: MCMC methods remain the most reliable methods for estimating the posterior distributions of uncertain model parameters and states. They, however, also remain computationally expensive. Ensemble Kalman filters provide computationally affordable approximations but rely on the assumptions of Gaussian probabilities. For nonlinear models even if the prior is Gaussian the posterior is not Gaussian anymore. Particle filtering on the other hand does not have such an assumption but requires a resampling step, which is usually stochastic. Ensemble transform particle filter is a particle filtering method that deterministically resamples the particles based on their importance weights and covariance maximization among the particles.

ETPF certainly outperforms ETKF for a one parameter nonlinear test case by giving a better posterior estimation. This conclusion also holds for the five parameter test case, however demands a substantially larger ensemble size. Moreover the mean estimations obtained by ETPF are not consistently better than the ones obtained by ETKF. When the number of uncertain parameters is large (2500) a decrease of degrees of freedom is essential. This is performed by localization. At large ensemble sizes ETPF performs as well as ETKF, while at small ensemble sizes ETKF still outperforms ETPF. Even though localized ETPF overfits the data less often than non-localized, localization destroys the property of ETPF to retain the imposed bounds. This results in deterioration of the first mode posterior approximation. Another approach to improve ETPF performance is instead of applying localization to use only first modes in the approximation of log permeabilty as they are better estimated by the method. An advantage of this approach is that it is fully Bayesian. However, one needs to know at which mode to make a truncation and this is highly dependent on the covariance matrix of the log permeability.

Reply to Referee 2

This article presents a comparison between an EnKF and a particle filter based approaches for parameter estimation in a time independent model. I think that this comparison is relevant and can provide good insights about the

performance of these two approaches in the context of parameter estimation. However I found many aspects that needs further clarification to support the conclusions made by the authors of this work. Major revisions are required to the paper.

Major points

- It is not clear if Importance Sampling and particle filters can be treated as synonyms. From my point of view particle filters can include different approaches for particle resampling to avoid the collapse of the filter and this is different from Importance Sampling which in principle does not include the resampling step.

Reply: Agree. We have changed the text accordingly.

-Page 2, 30 it is stated that particle filters do not update the uncertain parameters. This is not correct, many particle filters with different resampling approaches has been developed. These resampling steps introduce changes in the uncertain parameters so they get closer to the ones that produce the maximum observation likelihood, so the parameter ensemble evolves with time. It is true that the proposed technique performs this in a different way introducing a deterministic update of the parameter values (while usually resampling techniques in particle filters are stochastic). The difference between the implemented technique and previous techniques should be more clearly stated.

Reply: Agree. We have changed the text accordingly.

- In this work the implemented techniques are described as smoothers, however all the experiments performed are time independent. It is not clear for me what would be the difference between a filter or a smoother if there is no time involved. Please clarify this point. In the methodology I cannot find a difference between the filter implementation or the smoother implementation since there are no time index in the equations.

Reply: Agree, they are filters not smoothers. We have changed the text accordingly.

-Page 3, near 5: it is stated that ETKS does not employ the correlation in the estimation of the parameter. Filter equations are solved in the space defined by the ensemble members, but this implementation is basically equivalent to other EnKF which relies on the correlation between uncertain parameters and observed variables. Please clarify this point.

Reply: We have removed this sentence and also an iterative Kalman Smoother to avoid confusion. Both ETKF and ETPF considered in the paper solve equations defined in ensemble phase space.

-Page 8, 5 an iterative Kalman Smoother is mentioned here and shown in Figure 1, but detailed information about this technique is lacking. I suggest removing this technique since it has not been used in the experiments with the Darcy flow and also it has not been described in detail in the methodology section.

Reply: Agree, removed.

-In Figure 1, d, e and f a Gaussian prior produces a non-Gaussian posterior using ETKS. Since the EnKF relies on the linear and Gaussian assumption is it possible to obtain a non-Gaussian posterior from a Gaussian prior?

Reply: ETKF is able to give a non-Gaussian posterior due to the nonlinearity of the map between the uncertain parameters and observations.

-What is the motivation behind the functional introduced to define the observations in page 9, 15? What is $r_l$ which appears in the definition of $L_l(P)$?

Reply: $r_l$ denotes the location of the observation. This form of the observation functional and parameterization of the uncertain parameters given below guaranty the continuity of the forward map from the uncertain parameters to the observations and thus the existence of the posterior distribution as shown by Iglesias, M. A., Lin, K., and Stuart, A. M.: Well-posed Bayesian geometric inverse problems arising in subsurface flow, inverse problems, 30, 114 001, 2014. (This text in added to the revised version.)

-Figure 6 shows the distribution for the first 3 modes of Z. Please clarify how these modes are obtained.

Reply: For the log permeability we use Karhunen-Loeve expansions of the form

$$\log(k(x)) = \log(5) + \sum_{i=1}^{n^2} \sqrt{\lambda_i} \nu_i(x) \mathcal{Z}_i,$$

where $\lambda$ and $\nu(x)$ are eigenvalues and eigenfunctions of $\mathbf{C}$, respectively, and the vector $\mathcal{Z}$ is of dimension $n^2$ iid from a Gaussian distribution with zero mean and variance one. Making sure that the eigenvalues are sorted in descending order $\mathcal{Z}_i \sim \mathcal{N}(0,1)$ produces $\log(\mathbf{k}) \sim \mathcal{N}(\log(\mathbf{5}), \mathbf{C})$. (This text in added to the revised version.)

-Figure 8 shows that the RMSE associated with ETKS is always lower than the RMSE for ETPS, however the first 3 moments of Z are better estimated by ETPS than for ETKS. Does this mean that ETKS provides a better estimation of higher order modes?

Reply: The first three moments were averaged over 10 simulations and thus it was misleading to show and draw conclusions based on that figure. Instead we now plot a figure that shows an error of first three moments. We observe that in terms of the estimation of the first three modes ETPF outperforms ETKF. We explore the estimation based on only those modes further. We use only first three modes in the Karhunen-Loeve expansion when computing the estimated log permeability keeping the number of uncertain parameters the same, namely 2500. In Fig. 12 we observe that ETPF outperforms ETKF for large ensemble sizes independent of an initial sample. Moreover, ETPF is not overfitting the data anymore since RMSE always decreases after data assimilation except at small ensemble sizes. In Fig. 13 we show the mean fields for the best and worst

initial samples of $10^4$ size. ETPF gives RMSE at the best sample 31.1 and the worst sample 32.98. By comparing it to 30.51 and 39.2 obtained using the full Karhunen-Loeve expansions, we observe that the maximum RMSE over simulations decreased substantially, while the minimum RMSE only slightly increased. ETKF gives RMSE at the best sample 32.27 and the worst sample 33.23. (Compare to 32.48 and 33.9 using the full Karhunen-Loeve expansions). Thus ETKF slightly decreases both maximum and minimum RMSE over simulations.

-IS and ETKS provide spatially smother solutions than ETPS (Figure 10), however ETPS seems to provide a better representation of the spatial variability and patters of the parameter. The explanation provided by the authors is not convincing for me. IS with a large number of particles should provide a very good estimation of the parameters (this approach is used as a benchmark by the authors).
Reply: The spacial variability is indeed a result of sampling error. (This text in added to the revised version.)

-Also the distribution for the first 3 moments of Z are relatively similar between ETPS, ETKS and IS (but the spatial variability shown in Figure 10 are very different). This point is very important and I think it should be explored and discussed in more detail.
Reply: The distributions shown in Figure 6 (Fig. 11 in the revised version) are different between ETPF, ETKF and IS.

-The authors show that in many cases ETPS improves the fitting to the observations but degrades the RMSE of the parameter. Can this be due to an over fitting of the observations?
Reply: It is indeed due to an overfitting of the observations.

-For the experiments including localization, the authors do not show the spatial distribution of the estimated parameters. This is very important since using localization can significantly improve the small scale details in the estimated parameter field. This figure should be included in order to better evaluate the impact of localization.
Reply: New figures: Figure 15 (mean field) and Figure 16 (variance) are added to the revised version.

-It is also strange that there is almost no improvement between the global and local implementation of the ETKS algorithm. With such a large number of variables and for the smaller ensemble sizes a larger positive impact would be usually expected. The degradation of the ETKS with a small ensemble size using localization is unexpected. The authors indicate that better localization approaches should be used but previous studies usually indicates that the impact of localization is stronger for smaller ensemble sizes. Are other works that shows this kind of behavior with localization degrading the performance of the filter for small ensemble sizes?

Reply: At small ensemble sizes ETKF is less robust than at large ensemble sizes due to the sampling error and the localization radius was chosen based on 1 simulation and fixed for the remaining 9, which should not have been done. In the revised version the optimal localization radius was obtained over all 10 simulations. The results are shown in Fig. 12. At small ensemble sizes both ETKF and ETPF with localization give smaller misfit and RMSE and larger variance than without localization but ETKF still outperforms ETPF. For large ensemble sizes ETPF performs now comparably to ETKF. Moreover, for ensemble sizes greater than 150 all simulations result in the RMSE decrease after data assimilation (not shown).

-Page 16, before 5, it is stated that "However, IS does not change the parameters, only their weights, while ETPS does change the parameters. Therefore ETPS has an advantage of IS representing the correct posterior but does not have its disadvantage of resampling lacking". If the posterior is correct and tacking into account that there is no time evolution in this context, what would be the problem with the lacking of resampling in the IS? The results described in this section also suggest that the solutions provided by IS and ETPS are very similar given that the initial condition is the same (once again resampling does not seems to be an issue in this context).
Reply: This sentence is removed.

-Does ETPS with $10^5$ ensemble members produce a smooth field like the one produced by IS? In other words, the spatial variability that we see in Figure 10 b is produced by sampling errors or is the result of a better estimation of the parameter field? Results mentioned in the previous comment suggests that spatial variability is just a result of sampling noise and because of that is extremely sensitive to the prior ensemble. If we have a "lucky" prior then we end up with good results, but if the prior is bad then the result is also bad. In this sense ETKF seems to be more robust (which is reasonable when we need to update a large number of parameters with a relatively small ensemble and when the posterior distribution is not too far from a Gaussian).
Reply: Agree, the spatial variability that we see in Figure 10 b is indeed produced by sampling errors.

-Conclusions, page 19, 5: It is stated that ETPS better fit the posterior. However if we look at Figure 6 we found that for $10^4$ particles (which is a large ensemble for most applications), ETPS fit is very noisy. Can the authors perform and objective comparison between the posterior provided by IS and the posterior provided by ETPS and ETKS (for instance using the Kullback-Leibler divergence or other objective comparison between two distributions).
Reply: In order to perform an objective comparison between the probabilities we compute the Kullback-Leibler divergence of a posterior $\pi$ obtained by either ETPF or ETKF and the posterior $\pi^{\text{IS}}$ obtained by IS. ETPF gives the Kullback-Leibler divergence 0.21, 0.42, and 0.6, while ETKF 0.16, 0.07, and

0.49 for the modes $\mathcal{Z}_1$, $\mathcal{Z}_2$, and $\mathcal{Z}_3$, respectively. Thus ETKF gives a better approximation of the true pdf.

-Conclusions: Conclusions are very optimistic with respect to the performance of ETPS, however the RMSE of ETKS is always better in the large parameter space experiments. This suggests that the mean of the posterior is better estimated by ETKS rather than ETPS. While the mean is usually used as the best estimator of the parameter value, this should be mentioned in the conclusions.

Reply: ETPF certainly outperforms ETKF for a one parameter nonlinear test case by giving a better posterior estimation. This conclusion also holds for the five parameter test case, however demands a substantially larger ensemble size. Moreover the mean estimations obtained by ETPF are not consistently better than the ones obtained by ETKF. When the number of uncertain parameters is large (2500) a decrease of degrees of freedom is essential. This is performed by using localization. At large ensemble sizes (greater than 50) ETPF performs as well as ETKF, while at a small ensemble size of 10 ETKF still outperforms ETPF. Even though localized ETPF overfits the data less often than non-localized, localization destroys the property of ETPF to retain the imposed bounds. This results in deterioration of the first mode posterior approximation. Another approach to improve ETPF performance is instead of applying localization to use only first modes in the approximation of log permeabilty as they are better estimated by the method. An advantage of this approach is that it is fully Bayesian. However, one needs to know at which mode to make a truncation and this is highly dependent on the covariance matrix of the log permeability.

Minor points

Page 12, 5: It is stated that is assumed to be an exponential correlation with maximum correlation along 3pi/4 ... It is not clear for me the meaning of this sentence.

Reply: We removed this sentence.

Page 7, 20: It is stated that R0 approximation is used with large ensembles in the experiments presented in this work, but in the result section it is not clear if this approximation has been used or not.

Reply: We removed this approximation for consistency.

Figure 10, It would be nice to include grid lines or to include the observation location in all the panels just to have a reference to compare smaller scale details in the estimated parameters.

Reply: The observation locations are added to the plots.

[revised manuscript text omitted]

$$0 < b_{\min} \le b_m^b \le b_{\max} < 1, \quad m = 1, \dots, M,$$
$$0 < c_{\min} \le c_m^b \le c_{\max} < 1, \quad m = 1, \dots, M.$$

15  Now we assume two discrete random variables $B_1$ and $B_2$ have probability distributions given by

$$\pi(B_1 = \boldsymbol{u}_m^b) = 1/M, \quad \pi(B_2 = \boldsymbol{u}_m^b) = w_m^a,$$

with $w_m^a \ge 0$, $m = 1, \dots, M$ and $\sum_{m=1}^{M} w_m^a = 1$. As  ETPF looks for a matrix $\mathbf{T}^*$ which defines coupling between these two probability distributions, each entry of this coupling matrix satisfies the conditions given by Eq. (4)–(6). These conditions assure that each entry of the coupling matrix will be non-negative and less than 1. Since the analysis given by Eq. (8) is

20  $$\boldsymbol{u}_m^a = \begin{bmatrix} a_1^b(Mt_{1m}^*) + a_2^b(Mt_{2m}^*) + \cdots + a_M^b(Mt_{Mm}^*) \\ b_1^b(Mt_{1m}^*) + b_2^b(Mt_{2m}^*) + \cdots + b_M^b(Mt_{Mm}^*) \\ c_1^b(Mt_{1m}^*) + c_2^b(Mt_{2m}^*) + \cdots + c_M^b(Mt_{Mm}^*) \end{bmatrix}, \quad m = 1, \dots, M,$$

these conditions lead to

$$0 < a_{\min} \le a_m^a \le a_{\max} < 1, \quad m = 1, \dots, M,$$
$$0 < b_{\min} \le b_m^a \le b_{\max} < 1, \quad m = 1, \dots, M,$$

[revised manuscript text omitted]

10 them, which indicates that there is not enough spread in the ensemble. While for the parameter $c$ pdf shown in Fig. 3(h) this is an advantage of  ETKF, for the parameter $\log(k_2)$ pdf shown in Fig. 3(l) it is most certainly a disadvantage.  ETKF optimises for the mean (and variance), which is better approximated by  ETKF than by ETPF, as seen in Fig. 4(e). However this comes at a price of incorrect posterior shown in Fig. 3(k–l).

    In order to avoid any bias due to an initial ensemble we perform 10 simulations based on a random draw of an initial

15 ensemble from the same prior distributions. We conduct the numerical experiments for ensemble sizes varying from 10 to $10^3$ with an increment of 50. In  Fig. 4 we plot the true parameters $\boldsymbol{u}^{\text{true}}$, the mean $\bar{\bar{\boldsymbol{u}}}^a$ and the spread $\bar{\bar{\boldsymbol{u}}}^a \pm \bar{\boldsymbol{u}}^a_{\text{std}}$ of estimated parameters averaged over 10 simulations

$$\bar{\bar{u}}^a_i = \frac{1}{10}\sum_{r=1}^{10}\bar{u}^{a,r}_i, \quad \bar{u}^a_{\text{std}} = \frac{1}{10}\sum_{r=1}^{10}\sqrt{\frac{1}{M-1}\sum_{m=1}^{M}(u^{a,r}_{i,m} - \bar{u}^{a,r}_i)^2}, \text{ where } \bar{u}^{a,r}_i = \frac{1}{M}\sum_{m=1}^{M}u^{a,r}_{i,m}, \ r = 1,\dots,10,$$

$M$ is ensemble size, $i = 1,\dots,5$ is parameter index, and the superscript $a$ is for the analysis. We observe that both data

20 assimilation methods perform comparably in terms of mean estimation. The spread from  ETPF is, however, smaller than from  ETKF for each parameter. ETPF provides ensemble members that stay within the original bounds, while ETKF—outside the bounds.

[revised manuscript text omitted]

$$D_{KL}(\pi^{IS} \parallel \pi) = \sum_{i=1}^{N_b} \pi^{IS}(u_i) \log \frac{\pi^{IS}(u_i)}{\pi(u_i)},$$

where $N_b = 100$ is the number of bins. ETPF gives the Kullback-Leibler divergence 0.21, 0.42, and 0.6, while ETKF 0.16, 0.07, and 0.5 for the modes $\mathcal{Z}_1$, $\mathcal{Z}_2$, and $\mathcal{Z}_3$, respectively. Thus ETKF gives a better approximation of the true pdf. We use only first three modes in the Karhunen-Loeve expansion given by Eq. (14) when computing the estimated log permeability keeping the number of uncertain parameters the same, namely 2500. In Fig. 12(a) we observe that ETPF outperforms ETKF for large ensemble sizes independent of an initial sample. Moreover, ETPF is not overfitting the data anymore since RMSE

[Figure]

**Figure 10.** Squared error between the true and the mean estimated modes for $\mathcal{Z}_1$ (a), $\mathcal{Z}_2$ (b), and $\mathcal{Z}_3$ (c) w.r.t ensemble size. ETPF is shown in blue and ETKF in red with solid lines for median and shaded area for 25 and 75 percentile over 10 simulations. IS with ensemble size $10^5$ is in black.

**Table 1.** Optimal localization radius for ETPS and ETKS at different ensemble sizes M.

| M | 10 | 110 | 210 | ... | 910 |
|------|---------|-----|---------|-----|---------|
| ETPS |  0.4 | 0.6 |  0.6 | ... |  0.6 |
| ETKS |  0.6 | 1.2 | 1.2 | ... | 1.2 |

always decreases after data assimilation except at small ensemble sizes shown in Fig. 12(b). In Fig. 13 we show the mean fields for the best and worst initial samples of $10^4$ size. ETPF gives RMSE at the best sample 31.1 and the worst sample 32.98. By comparing it to 30.51 and  39.2 obtained using the full Karhunen-Loeve expansions, we observe that the maximum RMSE over simulations decreased substantially, while the minimum RMSE only slightly increased. ETKF gives RMSE at the best sample 32.27 and the worst sample 33.23. (Compare to 32.48 and 33.9 using the full Karhunen-Loeve expansions). Thus ETKF slightly decreases both maximum and minimum RMSE over simulations.

Next we employ localization for both  ETPF and ETKF. The optimal localization radius was obtained in terms of the smallest RMSE and shown in Table 1. It should be noted that smaller localization radius for ETPF than for ETKF was also observed

[Figure]

**Figure 11.** The posterior probability density function of parameters $\mathcal{Z}_1$ (left), $\mathcal{Z}_2$ (center), and $\mathcal{Z}_3$ (right). The posterior obtained by IS with ensemble size $10^6$ is plotted as a black line and the true parameter as a black cross. The posterior of ETPF is shown at the top and the posterior of ETKF at the bottom. Both ETPF and ETKF used $10^4$ ensemble members. The Kullback-Leibler divergence is in brackets.

by Chen and Reich (2015) for Lorenz 96 model and it is probably related to more noisy approximation of the posterior by ETPF than by ETKF. In Fig.  14 we plot misfit, RMSE and  variance.

5    At small ensemble sizes both ETKF and ETPF with localization give smaller misfit and RMSE and larger variance than without localization but ETKF still outperforms ETPF. For large ensemble sizes  ETPF performs now comparably to

10  ETKF (by increasing the localization radius to 1.2 we do not see an improvement in ETKF). Moreover, localized ETPF overfits the data less often than non-localized: $40\%$ agains $90\%$ for ensemble size 10 and $0\%$ agains non-zero% for ensemble sizes greater than  150 (not shown).

[Figure]

[Figure]

**Figure 12.**  Using only three first modes in the  KL expansion. Panel (a) : RMSE after data assimilation w.r.t ensemble size  with  mean, minimum and  maximum over 10 simulations for ETPF shown in blue and ETKF in red.  Panel (b) : % of simulations that result in $(RMSE^a - RMSE^b) > 0$ for ETPF.

In Fig. 15–16 we plot mean and variance of the log permeability field at ensemble size $10^3$ for ETPF (b)–(e) and ETKF (c)–(f) with localization at the smallest RMSE (b)–(c) and largest RMSE (e)–(f) over simulations, which are 32.29 and 34.08 for ETPF and 32.92 and 34.09 for ETKF, respectively. We observe that localization decreases the sampling noise and the spatial variability of the mean field obtained by ETPF at ensemble size $10^3$ resembles IS at ensemble size $10^5$. The variance obtained by ETPF with localization  16(b–e) has also improved.

The posterior estimation of the first mode $\mathcal{Z}_1$, however, degraded , while of $\mathcal{Z}_2$ and $\mathcal{Z}_3$ improved. The Kullback-Leibler divergence for the first mode is 0.73 (compare to 0.21 without localization), and for second and third is 0.2 and 0.18, correspondently (compare to 0.42 and 0.6 without localization). Variance of the posteriors is larger when localization is applied for both ETPF and ETKF. The localized weights given by Eq. (11) vary less than the non-localized weights given by Eq. (3). Therefore the  localized pdf is less noisy than the non-localized. However, localization applied in the form of the Karhunen-Loeve expansion given by Eq. (14) does not retain the imposed bounds on the modes $\mathcal{Z}$ as we need to invert a

[Figure]

**Figure 13.** Same as figure 8,  but using only three first modes in the KL expansion.

matrix product of eigenvalue and eigenvector matrices to obtain the modes. By increasing the localization radius to 1.2 we get the Kullback-Leibler divergence 0.64 for the first mode, and 0.13 and 0.11 for the second and third, correspondently, thus the posterior approximation improves only slightly.

**5 Conclusions**

[revised manuscript text omitted]

---

## Author Response (AR2)

Reply to Referee 1

[Major Comments]
1. I wonder why ETPF can produce finer patterns of permeability field than ETKF (Fig. 8). Also, why does localization lead to spatially-coarser patterns (Fig. 15)? In the atmospheric data assimilation, the localization generally enables to obtain spatially finer patterns in analyses.
Reply: ETPF is affected by sampling noise at small scales more than at large scales, thus the fine pattern of permeability field shown in Fig. 8. When localization is applied it smooths out the unobserved small scales, thus the spatially-coarse pattern of permeability field shown in Fig. 15.

2. I became to wonder whether or the RMSE is a good measure to evaluate estimated fields. As shown in the results by ETKF, estimating spatially-smoothed fields would prevent terrible scores in RMSE. If authors can evaluate with an additional metric, it would be beneficial.
Reply: Indeed the RMSE alone is not a good measure to evaluate estimated fields for the above reasons. That is why in addition to RMSE we studied the posterior of modes and used the Kullback-Leibler divergence to assess the posterior estimations.

3. Is it allowable to tune the localization scale not for data misfits but for RMSE? If we think about realistic applications, such tuning of localization is not affordable.
Reply: We agree that for realistic applications, it is indeed unaffordable. However, it is a standard set-up for theoretical investigation of a data assimilation method.

4. I recommend revising Fig. 14 to contain two experiments (with and without localization) directly. It would be possible to remove Min/Max since they are not discussed here.
Reply: Agree.

5. Readers would be interested a lot in how to choose ETKF or ETPF, and with or without localization. If authors could give some strategies (inferences), it would be very beneficial. Also, if there would be limitations or issues for realistic applications, please discuss them in the last section.
Reply: We believe that ETPF is promising for the inverse modelling. However, more theoretical studies have to be performed for ETPF before it is considered for realistic applications. Plausible issues related to realistic application are numerous accurate observations, time-dependency of an underlined model, and flow being multiphase, for example.

6. Title of the manuscript seems too broad
Reply: Changed to "Application of ensemble transform data assimilation methods for parameter estimation in reservoir modelling"

[Minor Comments]

1. Please revise the caption of Fig. 8.
Reply: Agree. Changed to "Log permeability field with dots representing the observation locations. Truth is shown in (a) and mean obtained by IS with ensemble size $10^5$ in (d). Mean obtained with ensemble size $10^3$ by ETPF shown in (b–e) and by ETKF in (c–f), where (b–c) are at the smallest RMSE and (e–f) are at the largest RMSE over simulations. The corresponding RMSE is given in brackets."

Reply to Referee 2

General comment
One important point that needs to be further clarified is what is the motivation to use ETPF in comparison with the classical IS for this particular application.
Reply: The drawback of IS is that it does not update the uncertain parameters but only their weight, thus a computationally unaffordable ensemble is required. In order to decrease this cost a family of particle filters has been developed where IS is with resampling and a sample is called particle. The resampling in particle filtering is, however, stochastic. Ensemble Transform Particle Filter (ETPF) is a particle filtering method that deterministically resamples the particles.

Minor points

Page 1, line 19 due to sampling error.
Reply: Agree.

Page 1, line 24 leading modes instead of "first modes"?
Reply: Agree.

Page 1, line 24 This approach is ... this sentence is not clear please rephrase it.
Reply: Agree. Changed to "which however demands a knowledge at which mode to truncate."

Page 7, line 4 introduce sampling errors ?
Reply: Agree.

Page 11, line 10 The sentence starting with While for the parameter c ... is not clear. Please rephrase it.
Reply: Agree. Removed.

Page 11, line 13, Figure 4 is introduced but the quantities shown here are disused in line 18 of the same page.
Reply: Agree. Removed.

Page 11, line 14. In order to avoid any bias... may be better to say in order to increase the robustness of the results or in order to filter the sensitivity of the results to the initial parameter ensemble.
Reply: Agree. Changed to "to check the sensitivity of the results to the initial parameter ensemble..."

Page 11, line 20. Is there a way to know which method produce the best spread? Can the spread be compared to the estimation error to measure which method produce the best error / spread relationship.
Reply: Even though both methods are slightly underdispersive, as the spread to error ratio is below 1, ETKF gives better ratio for all parameters but $\log(k_2)$ as seen in Figure 1 of the response to the reviewers.

[Figure]

Figure 1: Spread to error ratio w.r.t ensemble size. ETPF is on the left and ETKF is on the right.

Figure 4: It seems that both methods (ETKF and ETPF) have a bias in the estimation of parameters a, b and c. Is there a reason for that? Does IS also show that bias?
Reply: We observe that all methods including IS have bias in the estimations of geometrical parameters, which is due to the small number of observations. ETPF and ETKF perform comparably in terms of mean estimation, though some are better estimated by ETKF and other are better estimated by ETPF. Comparing the error in pressure of the mean parameters seen in Fig. 2 of the response to the reviewers, we observe that methods are equivalent, which is a manifestation of the ill-posedness of the problem.

Page 11, line 21. The behaviour of the estimated parameters with respect to their bounds is not clearly seen from Figs 3 or 4.
Reply: Agree. Removed.

[Figure]

Figure 2: RMSE of pressure w.r.t ensemble size.

Page 12, It would also be interesting to see which parameter set produce a better estimation of the true pressure field. Parameter comparison alone is not conclusive since some parameters are better estimated by the ETKF and others better estimation by the ETPF, computing the error in P will help to decide which estimated parameter set produce the best results.
Reply: Agree, please see reply above.

Figure 5 I suggest removing the 0 line from the figure legend.
Reply: Agree.

Page 16, line 13, this means that ETPF sensitivity to the initial ensemble is due to the ...
Reply: Agree.

Page 16, line 13. If IS produce the same result with the same ensemble size, what is the advantage of the ETPS in this application? It is important to provide a good motivation (which can be verified with the results) for using a more complex and computationally expensive technique.
Reply: Agree. Removed.

Page 16, line 17. In Fig. 9 we plot the variance ...
Reply: Agree.

Page 16, line 19. Why ETKF provides a smother variance? Is the variance estimation provided by the ETKF better or worse than the one provided by ETPF?
Reply: ETKF provides smoother variance than ETPF due to smaller sampling errors.

Figure 13, it seems that ETPF is more affected by sampling noise at small scales, so using a truncated representation of the fields significantly improve the results for ETPS. ETKF is more robust, it is probably filtering out the small scales (which are poorly observed and more difficult to retrieve) and because of this its performance is less affected by the truncation presented in this figure.
Reply: Agree. Added.

Table 1: In this table ETPS and ETKS are used instead of ETPF and ETKF. Note also that the optimal localization radius do not change much with ensemble size (usually in the ETKF the localization radius increases with the increase of the ensemble size eventually going to infinity).
Reply: The optimal localization radius between 0.2 and 1.2 was obtained in terms of the smallest RMSE, when the domain is 1 by 1.

Page 20, line 10. The sentence between parenthesis is not clear. How do the authors select the localization radius for each method?
Reply: The optimal localization radius between 0.2 and 1.2 was obtained in terms of the smallest RMSE. However, we also checked the performance of non-optimal LETPF. We removed this sentence as no results are discussed anyway.

For clarity the localized versions of ETPF and ETKF can be referred as LETPF and LETKF respectively.
Reply: Agree.

Page 21, line 16. However , localization applied in the form of the ... I think that as long as the values of the parameters are within the required range the method achieves something interesting for parameter estimation (since parameter bounds are usually difficult to handle with other techniques like the EnKF).
Reply: It should be noted that unlike ETKF, LETPF does not converge to ETPF as the localization radius goes to infinity due to the transport problem being univariate for LETPF and multivariate for ETPF.

Page 24, line 10. This conclusion also holds for the five parameter test case. From my point of view the results are not so conclusive in this case. An objective measure of the posterior error is missing in the discussion of the results.
Reply: For the five parameter test case, the mean estimations obtained by ETPF are not consistently better than the ones obtained by ETKF and the

spread is smaller. The Kullback-Leibler divergence from ETKF is smaller than from ETPF for all parameters. (We added analysis of the Kullback-Leibler divergence in the five-parameter section.)

Page 24, line 7. Another approach ... This approach also assumes that the log permeability can be represented as a Gaussian process. It is also not clear for me why the authors state that this approach is fully Bayesian?

[revised manuscript text omitted]
{y}^{a,r} - y_{\text{obs}})^T R^{-1}(\bar{y}^{a,r} - y_{\text{obs}}), \ r = 1,\dots,10 \tag{13}$$

after data assimilation. The same metrics are computed before data assimilation and denoted by a superscript $b$. In Fig. 5(a)–(b) we plot $(\text{misfit}^{a,r} - \text{misfit}^{b,r})$ and $(\text{RE}^{a,r} - \text{RE}^{b,r})$, respectively, for each simulation $r$ as a function of ensemble size. ETPF is

[Figure]

**Figure 4.** $\bar{\bar{\boldsymbol{u}}}^a$ and $\bar{\bar{\boldsymbol{u}}}^a \pm \bar{\boldsymbol{u}}^a_{\text{std}}$ w.r.t ensemble size: (a) for the parameter $a$, (b) for $b$, (c) for $c$, (d) for $\log(k_1)$, (e) for $\log(k_2)$. ETPF is shown in blue, ETKF in red, the true parameters are in black and the mean of IS in magenta.

shown in blue and ETKF in red. Black line is at zero level. Positive values of the differences mean an increase of either data mismatch or relative error after data assimilation. We observe a data misfit decrease for both ETPF and ETKF except at an ensemble size 10. RE does not always decrease for ETPF: for some simulations ETPF is at zero level or slightly above it, while for ETKF the sole exception is at an ensemble size 10.

**4.2 High-dimensional nonlinear problem**

Next, we consider a high-dimensional problem where the dimension of the uncertain parameter is $n^2 = 2500$. The domain $D$ is now not divided into subdomains. However, unlike in the previous test case here we implement a spatially varying permeability field. We assume the log permeability is generated by a random draw from a Gaussian distribution $\mathcal{N}(\log(\mathbf{5}), \mathbf{C})$. Here $\mathbf{5}$ is an $n^2$ vector with all 5. $\mathbf{C}$ is assumed to be an exponential correlation with an element of $\mathbf{C}$ being

$$C_{i,j} = \exp(-3(|h_{i,j}|/v)), \; i,j = 1,\ldots,n^2.$$

[Figure]

**Figure 5.** $\mathrm{misfit}^{a,r} - \mathrm{misfit}^{b,r}$ (a) and $\mathrm{RE}^{a,r} - \mathrm{
[revised manuscript text omitted]

---

## Author Response (AR3)

We apologise to the Editor for not making clear the changes in the response to the reviewers.

Reply to Referee 1

[Major Comments]

1. I wonder why ETPF can produce finer patterns of permeability field than ETKF (Fig. 8). Also, why does localization lead to spatially-coarser patterns (Fig. 15)? In the atmospheric data assimilation, the localization generally enables to obtain spatially finer patterns in analyses.

Reply: ETPF is affected by sampling noise at small scales more than at large scales, thus the fine pattern of permeability field shown in Fig. 8. When localization is applied it smooths out the *unobserved* small scales, thus the spatially-coarse pattern of permeability field shown in Fig. 15.

Added on P.15 L.5: This means that ETPF sensitivity to the initial sample is due to sampling error and that the spatial variability of ETPF is a result of sampling error.

Added on P.15 L.31: ETPF is more affected by sampling noise at small scales, so using a truncated representation of the fields significantly improves the results for ETPF. ETKF is filtering out the small scales that are not observed and thus is less affected by the truncation.

2. I became to wonder whether or the RMSE is a good measure to evaluate estimated fields. As shown in the results by ETKF, estimating spatially-smoothed fields would prevent terrible scores in RMSE. If authors can evaluate with an additional metric, it would be beneficial.

Reply: Indeed the RMSE alone is not a good measure to evaluate estimated fields for the above reasons. That is why in addition to the RMSE we studied the posterior of modes and used the Kullback-Leibler divergence to assess the posterior estimations. We also added an analysis of the KL divergence for the 5-parameter test case in the revised version. Please see P.10 L.4–9.

3. Is it allowable to tune the localization scale not for data misfits but for RMSE? If we think about realistic applications, such tuning of localization is not affordable.

Reply: We agree that for realistic applications, it is indeed unaffordable. However, it is a standard set-up for a theoretical investigation of a data assimilation method.

4. I recommend revising Fig. 14 to contain two experiments (with and without localization) directly. It would be possible to remove Min/Max since they are not discussed here.

Reply: Agree. Please see revised Fig. 14.

5. Readers would be interested a lot in how to choose ETKF or ETPF, and with or without localization. If authors could give some strategies (inferences),

it would be very beneficial. Also, if there would be limitations or issues for realistic applications, please discuss them in the last section.

Reply: We have rewritten the conclusions taking into account the suggestions. Starting from P.18 L.13: Ensemble Kalman filters (ETKF) provide computationally affordable approximations but rely on the assumptions of Gaussian probabilities. For nonlinear models even if the prior is Gaussian the posterior is not Gaussian anymore. Particle filtering on the other hand does not have such an assumption... Another plausible drawback of localization is an assumption of observations being local, which might not be the case for inverse modelling. An alternative approach to improve ETPF performance is instead of applying localization to use only leading modes in the approximation of log permeability, as they are better estimated by the method. However, one needs to know at which mode to truncate and this is highly dependent on the covariance matrix of log permeability. To conclude, we believe that ETPF is promising for inverse modelling. However, more theoretical studies have to be performed for ETPF before it is considered for realistic applications. Plausible issues related to realistic application are numerous accurate observations, time-dependency of an underlying model, and a flow being multiphase, for example.

6. Title of the manuscript seems too broad

Reply: Changed to "Application of ensemble transform data assimilation methods for parameter estimation in reservoir modelling"

[Minor Comments]

1. Please revise the caption of Fig. 8.

Reply: Agree. Changed to "Log permeability field with dots representing the observation locations. Truth is shown in (a) and mean obtained by IS with ensemble size $10^5$ in (d). Mean obtained with ensemble size $10^3$ by ETPF shown in (b–e) and by ETKF in (c–f), where (b–c) are at the smallest RMSE and (e–f) are at the largest RMSE over simulations. The corresponding RMSE is given in brackets."

Reply to Referee 2

General comment

One important point that needs to be further clarified is what is the motivation to use ETPF in comparison with the classical IS for this particular application.

Reply: Added on P.2 L.23: The drawback of IS is that it does not update the uncertain parameters but only their weight, thus a computationally unaffordable ensemble is required. In order to decrease this cost a family of particle filters (Doucet et al., 2001) has been developed where IS is supplied with resampling, and a sample is called particle... The resampling in particle filtering is, however, stochastic. Ensemble Transform Particle Filter (ETPF) developed by Reich and Cotter (2015) is a particle filtering method that deterministically resamples the particles based on their weights and covariance maximization among the particles.

Minor points

Page 1, line 19 due to sampling error.
Reply: Agree. Changed on P.1 L.14: due to sampling error

Page 1, line 24 leading modes instead of "first modes"?
Reply: Agree. First modes changes to leading modes in the whole manuscript.

Page 1, line 24 This approach is ... this sentence is not clear please rephrase it.
Reply: Agree. Changed on P.1. L.19: which however demands a knowledge at which mode to truncate.

Page 7, line 4 introduce sampling errors ?
Reply: Agree. Changed on P.6 L.15: This limit of a small ensemble size introduces sampling errors.

Page 11, line 10 The sentence starting with While for the parameter c ... is not clear. Please rephrase it.
Reply: We removed this sentence.

Page 11, line 13, Figure 4 is introduced but the quantities shown here are disused in line 18 of the same page.
Reply: Agree. We now introduce the quantities on P.10 L.14. Then we discuss these quantities shown in Fig. 4.

Page 11, line 14. In order to avoid any bias... may be better to say in order to increase the robustness of the results or in order to filter the sensitivity of the results to the initial parameter ensemble.
Reply: Agree. Changed on P.10 L.10: to check the sensitivity of the results to the initial parameter ensemble...

Page 11, line 20. Is there a way to know which method produce the best spread? Can the spread be compared to the estimation error to measure which method produce the best error / spread relationship.
Reply: Added on P.10 L.20: Both methods are slightly underdispersive as the spread to error ratio is below 1. For ensemble size $10^3$ ETKF gives (0.95 0.88 0.88 0.97 0.98) and ETPF gives (0.92 0.81 0.84 0.99 0.86) for ($a$ $b$ $c$ $\log(k_1)$ $\log(k_2)$). Thus ETKF gives better ratio for all parameters but $\log(k_1)$.
We provide the reviewers with an additional plot of the spread to error ratio for all ensemble sizes (Figure 1 here) but omit it in the manuscript.

Figure 4: It seems that both methods (ETKF and ETPF) have a bias in the

[Figure]

Figure 1: Spread to error ratio w.r.t ensemble size. ETPF is on the left and ETKF is on the right.

estimation of parameters a, b and c. Is there a reason for that? Does IS also show that bias?
Reply: Added on P.10 L.15: We observe that all the methods including IS have a bias in the estimations of geometrical parameters, which is due to a small number of observations. Please also see revised Fig. 4.

Page 11, line 21. The behaviour of the estimated parameters with respect to their bounds is not clearly seen from Figs 3 or 4.
Reply: Agree. We removed the sentence about the bounds.

Page 12, It would also be interesting to see which parameter set produce a better estimation of the true pressure field. Parameter comparison alone is not conclusive since some parameters are better estimated by the ETKF and others better estimation by the ETPF, computing the error in P will help to decide which estimated parameter set produce the best results.
Reply: Agree. Added on P.10 L.16: ETPF and ETKF perform comparably in terms of the mean estimation, though some parameters are better estimated by ETKF and others are better estimated by ETPF. Comparing the error in the mean pressure we observe that the methods are equivalent (thus not shown in the manuscript but shown in the response to the reviewers in Fig. 2 here), which is a manifestation of the ill-posedness of the problem.

Figure 5 I suggest removing the 0 line from the figure legend.
Reply: Agree. Removed.

Page 16, line 13, this means that ETPF sensitivity to the initial ensemble is due to the ...
Reply: Agree. Added on P.15 L.5: This means that ETPF sensitivity to the initial sample is due to sampling error and that the spatial variability of ETPF

[Figure]

Figure 2: RMSE of pressure w.r.t ensemble size.

is a result of sampling error.

Page 16, line 13. If IS produce the same result with the same ensemble size, what is the advantage of the ETPS in this application? It is important to provide a good motivation (which can be verified with the results) for using a more complex and computationally expensive technique.
Reply: Agree. Added on P.15 L.6: It should be noted that IS with ensemble size $10^3$ and this good initial ensemble gives the RMSE 30.51 and the same mean log permeability field as ETPF shown in Fig. 8(b). However, IS does not change the parameters, only their weights, while ETPF does change the parameters. Therefore ETPF has an advantage of IS representing the correct posterior but does not have its disadvantage of resampling lacking.

Page 16, line 17. In Fig. 9 we plot the variance ...
Reply: Agree. Changed on P.15 L.10: In Fig. 9 we plot the variance...

Page 16, line 19. Why ETKF provides a smother variance? Is the variance estimation provided by the ETKF better or worse than the one provided by ETPF?
Reply: Added on P.15 L.12: ETKF provides smoother variance than ETPF due

to smaller sampling errors.

Figure 13, it seems that ETPF is more affected by sampling noise at small scales, so using a truncated representation of the fields significantly improve the results for ETPF. ETKF is more robust, it is probably filtering out the small scales (which are poorly observed and more difficult to retrieve) and because of this its performance is less affected by the truncation presented in this figure. Reply: Agree. Added on P.15 L.31: ETPF is more affected by sampling noise at small scales, so using a truncated representation of the fields significantly improves the results for ETPF. ETKF is filtering out the small scales that are not observed and thus is less affected by the truncation.

Table 1: In this table ETPS and ETKS are used instead of ETPF and ETKF. Note also that the optimal localization radius do not change much with ensemble size (usually in the ETKF the localization radius increases with the increase of the ensemble size eventually going to infinity). Reply: Added on P.15 L.34: The optimal localization radius between 0.2 and 1.2 was obtained in terms of the smallest RMSE.

Page 20, line 10. The sentence between parenthesis is not clear. How do the authors select the localization radius for each method? Reply: The optimal localization radius between 0.2 and 1.2 was obtained in terms of the smallest RMSE. However, we also checked the performance of non-optimal LETPF. However, we removed this sentence as no results are discussed anyway.

For clarity the localized versions of ETPF and ETKF can be referred as LETPF and LETKF respectively. Reply: Agree. We changed it in the whole manuscript.

Page 21, line 16. However , localization applied in the form of the ... I think that as long as the values of the parameters are within the required range the method achieves something interesting for parameter estimation (since parameter bounds are usually difficult to handle with other techniques like the EnKF). Reply: Added on P.18 L.9: ...unlike ETKF, LETPF does not converge to ETPF as the localization radius goes to infinity due to the transport problem being univariate for LETPF and multivariate for ETPF.

Page 24, line 10. This conclusion also holds for the five parameter test case. From my point of view the results are not so conclusive in this case. An objective measure of the posterior error is missing in the discussion of the results. Reply: We added analysis of the Kullback-Leibler divergence for the five-parameter test case as an objective measure of the posterior error, please see P.10 L.4–9. We change the conclusions on P.19 L.5: For the five parameter test case, the mean estimations obtained by ETPF are not consistently better than the ones obtained by ETKF and the spread is smaller. The Kullback-Leibler divergence

from ETKF is smaller than from ETPF for all the parameters.

Page 24, line 7. Another approach ... This approach also assumes that the log permeability can be represented as a Gaussian process. It is also not clear for me why the authors state that this approach is fully Bayesian?

[revised manuscript text omitted]